# FLOW: MODULARIZED AGENTIC WORKFLOW AUTOMATION

**Boye Niu**[1]**, Yiliao Song**[2]**, Kai Lian**[1]**, Yifan Shen**[4]**, Yu Yao**[1]**, Kun Zhang**[3,4]**, Tongliang Liu**[1,4†]

[1]Sydney AI Centre, The University of Sydney, [2]The University of Adelaide
[3]Carnegie Mellon University, [4]Mohamed bin Zayed University of Artificial Intelligence

## ABSTRACT

Multi-agent frameworks powered by large language models (LLMs) have demonstrated great success in automated planning and task execution. However, the effective adjustment of agentic workflows during execution has not been well studied. An effective workflow adjustment is crucial in real-world scenarios, as the initial plan must adjust to unforeseen challenges and changing conditions in real time to ensure the efficient execution of complex tasks. In this paper, we define *workflows as an activity-on-vertex (AOV) graph*, which allows continuous workflow refinement by LLM agents through dynamic subtask allocation adjustment based on historical performance and previous AOVs. To further enhance framework performance, we emphasize *modularity* in workflow design based on evaluating parallelism and dependency complexity. With this design, our proposed multi-agent framework achieves efficient concurrent execution of subtasks, effective goal achievement, and enhanced error tolerance. Empirical results across various practical tasks demonstrate significant improvements in the efficiency of multi-agent frameworks through dynamic workflow refinement and modularization. The code is available at: https://github.com/tmllab/2025_ICLR_FLOW.

## 1 INTRODUCTION

Large Language Models (LLMs) (Significant Gravitas; Zhou et al., 2023) show a remarkable ability to understand and generate human-like text. Recent advances have significantly enhanced their capability to emulate human reasoning (Sun et al., 2024), indicating a promising future for LLM-based reasoning. With the powerful ability to handle a variety of natural language processing tasks, these models underpin a wide range of applications, from conversational agents (Ye et al., 2024) and content creation tools (Yao et al., 2023) to advanced analytics and decision-making systems (Ramesh et al., 2021; Wang et al., 2023). Building upon this foundation, a key advancement is the development of ***multi-agent frameworks empowered by LLM*** (Liu et al., 2023; Li et al., 2023; Hong et al., 2024b; Wu et al., 2024; Wang et al., 2024; Chen et al., 2024; Liu et al., 2024) where multiple LLM-based agents collaborate to address complex tasks, leveraging their collective reasoning and planning abilities to automate and optimize task execution processes.

Existing LLM-based multi-agent frameworks define LLM as an agent, and agents collaborate with each other via manually designed or LLM-generated prompts. Specifically, *MetaGPT* focuses on programming tasks by leveraging Standardized Operating Procedures (SOPs) (Wooldridge & Jennings, 1998; DeMarco & Lister, 2013; Belbin, 2010). It predefined distinct roles such as product manager, project manager, and engineer. For each role, an LLM agent is initialized, and these agents operate within a strict and sequential workflow to execute subtasks. *CAMEL* can complete a variety of task types. It requires users to pre-define two agents. These agents interact and execute tasks sequentially, each agent taking on specific responsibilities. *AutoGen* is also aimed at completing diverse tasks. Unlike CAMEL, AutoGen automatically creates an agent list with different roles based on subtask requirements. These agents execute subtasks sequentially following the order in the list.

---

[†]Correspondence to Tongliang Liu (tongliang.liu@sydney.edu.au)

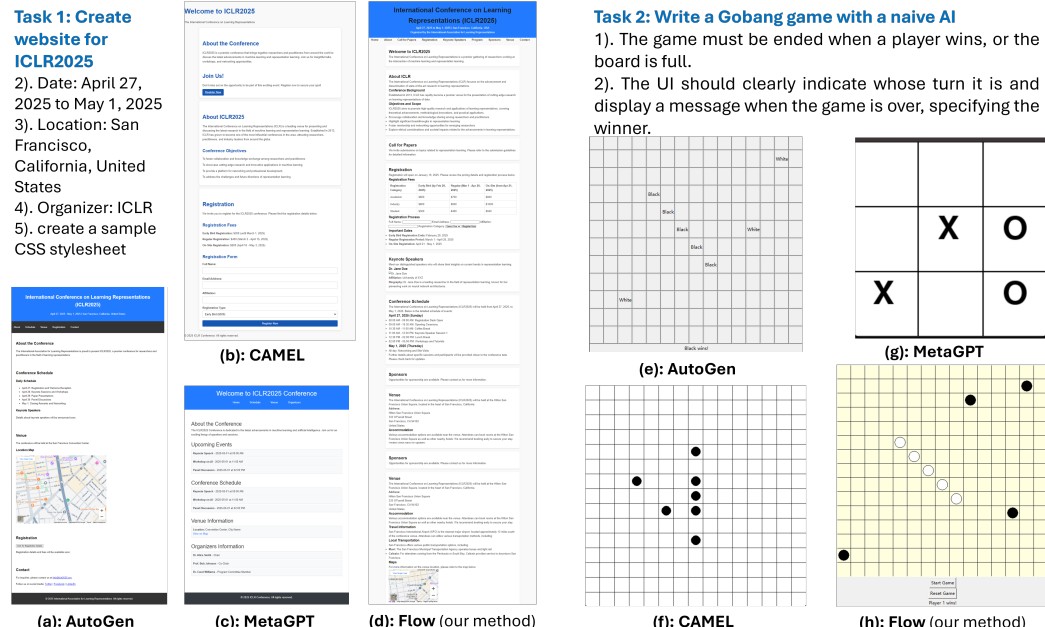

Figure 1: Comparative evaluations among four frameworks—AutoGen, CAMEL, MetaGPT, and Flow (ours)—across two tasks, present notable differences in performance. For the left task, AutoGen, CAMEL, and MetaGPT only managed to produce basic designs lacking in completeness while Flow excelled by creating a fully developed and well-structured website. For the right task, Flow demonstrated superior capability by successfully generating a working game with a clear and intuitive interface, while the other frameworks struggled to deliver fully functional code.

Building upon the strengths of current multi-agent frameworks, our work aims to further improve existing general-purpose multi-agent frameworks by enabling ***dynamically updating workflows*** during task execution and encouraging ***modularity*** in workflows when planning the workflows.

Specifically, ***dynamic updating workflows*** allow agents to adjust **subtask allocations** and **agent roles** in real-time based on ongoing performance feedback and changing conditions. This capability **ensures** that the system remains responsive and efficient even when faced with unexpected obstacles. For instance, if an agent encounters a roadblock in data preprocessing, the system can reassign this subtask to another agent or introduce a new subtask to resolve the issue. This adaptability is essential for maintaining robustness and ensuring seamless execution of complex tasks.

***Modularity*** in system design involves dividing a system into separate, independently operating modules, each responsible for specific functionalities (Baldwin & Clark, 1999). In *our context*, **modularity** ***refers to the decomposition of a complex task into smaller, interchangeable subtask modules***. A highly modularized workflow enables subtasks to execute concurrently, without bottlenecks from other parts of the workflow, and thereby directly improves the operational efficiency of multi-agent frameworks. Furthermore, **modularity** enhances the ease of dynamic updating. When workflows are highly modularized, the dependency complexity between subtasks is minimal. Therefore, updating one subtask does not affect others, allowing for small workflow adjustments. For example, if an agent responsible for data preprocessing encounters an unexpected obstacle, a ***system of high modularity*** can adapt by introducing only one subtask with minimal impact on the rest of the workflow.

In this paper, we enhance existing multi-agent frameworks by achieving modularity and enabling dynamic workflow updates. Our framework allows agents to execute their subtasks in parallel while facilitating efficient workflow updates. This is accomplished by formulating the entire workflow as an Activity-on-Vertex (AOV) graph, which is a directed acyclic graph (DAG) where each subtask is represented as a node with its status and generated logs, while the directed edges capture dependencies between subtasks. To encourage a modular workflow design from the beginning, we generate multiple candidate AOV graphs for the task. These candidates are then evaluated based on their degree of

parallelism and the complexity of their dependencies. The AOV graph with the highest parallelism and lowest dependency complexity is selected.

During task execution, our framework continuously checks and refines the workflow, updating it when a subtask fails (see Fig. 2: *Check & Refine*). The framework updates subtask allocations and agent roles based on ongoing performance data and the current workflow. As our AOV-based workflow encourages high modularity, updating one module does not necessarily affect others, allowing for localized adjustments during workflow updates (see Fig. 2: *Update*). Similar to the initial workflow generation, multiple AOV graphs are generated and the one with **highest parallelism** and **lowest dependency complexity** is selected during dynamic updates. This iterative workflow refinement process enhances adaptability to new challenges and evolving objectives throughout task execution, ensuring dynamic workflow updates without compromising overall performance.

**Contributions:** 1) We introduce and encourage modularity in multi-agent workflows, emphasizing the design of workflows with high parallelism and low dependency complexity. This modular design enhances efficiency, robustness, and scalability by enabling concurrent subtask execution and minimizing bottlenecks caused by complex interdependence. 2) We propose a practical multi-agent framework that supports highly flexible updates to the workflow during runtime. Our method enables local updates to the entire workflow based on global information, allowing agents to efficiently adapt to unexpected challenges while maintaining system coherence and consistency. 3) Through comprehensive experiments, we demonstrate significant improvements in both the adaptability and efficiency of our multi-agent framework compared to existing approaches.

## 2 RELATED WORK

**LLM-based Task Decision-Making.** Recent developments in LLM-based task decision making have focused on improving the reasoning and planning abilities of agents (Yao et al., 2023; Song et al., 2023; Zhou et al., 2024; Prasad et al., 2023; Shinn et al., 2023; Ahn et al., 2022; Teng et al., 2025). Previous approaches like ReAct (Yao et al., 2023) iteratively generate thoughts and actions based on current observations until task completion. This framework integrates action-taking with reasoning, allowing agents to perform complex tasks in dynamic environments. Reflexion (Shinn et al., 2023) further improves this by incorporating self-reflection, where the agent evaluates and adjusts its reasoning during execution. ADAPT (Prasad et al., 2023) introduces recursive task decomposition, enabling LLM-based agents to break tasks into smaller subtasks, which leads to improved task execution flexibility. However, these approaches often overlook dynamic task reallocation, particularly in multi-agent settings, which is where our work extends current research.

**LLM-based Multi-Agent Frameworks.** Multi-agent frameworks have long been employed for task execution in distributed environments, with recent advances leveraging LLM to enhance coordination and decision-making (Hong et al., 2024b; Li et al., 2023; Wu et al., 2024; Hu et al., 2024; Zhang et al., 2024; Trirat et al., 2024). However, existing frameworks often rely on static workflows with limited adaptability to changes in the task environment. DyLAN (Liu et al., 2024) and MACNET (Qian et al., 2024) utilize static graphs to represent workflows in multi-agent frameworks; GPTSwarm (Zhuge et al., 2024) enhances agent interactions but maintains a fixed agent topology; DataInterpreter (Hong et al., 2024a) updates workflows primarily in response to execution failures in subtasks, adjusting subsequent tasks while leaving completed tasks unchanged; AFlow (Zhang et al., 2025) introduces a dynamic workflow generation framework based on Monte Carlo Tree Search, enabling adaptive adjustments through iterative code modification. This highlights the need for dynamic workflow updates.

## 3 THE PROPOSED MULTI-AGENT FRAMEWORK: FLOW

Our proposed Flow enhances multi-agent frameworks powered by LLM by introducing modularity and dynamic workflow updating. As depicted in Fig. 2, given the task requirement, Flow first *formulates the initial workflow* for *execution plan generation and agent allocation*. During execution, the workflow is *continuously refined and dynamically updated* until the task is completed. To maximize system simplicity and flexibility, we design a dictionary-based structure for *implementation*. In the following, we detail how to achieve these features.

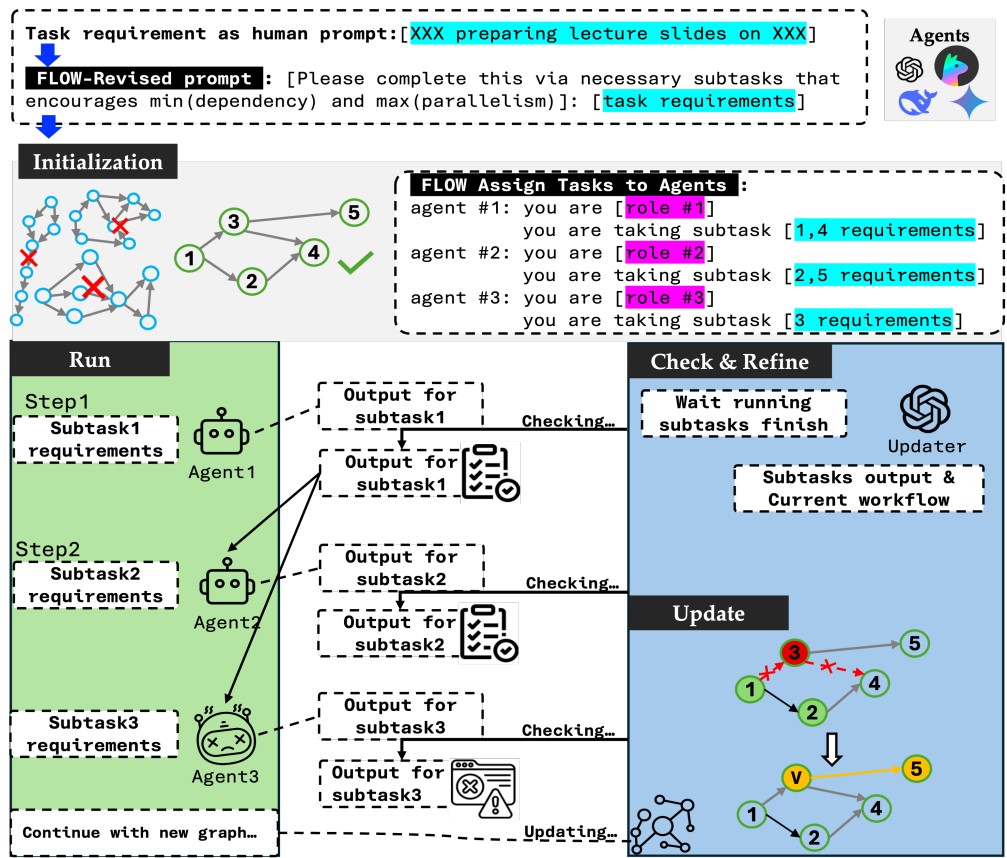

Figure 2: The process starts with task initialization, encouraging the modularity and execute parallel of subtasks. Outputs are evaluated. If errors are detected, the workflow is dynamically updated by modifying the task graph. This iterative process continues until successful task completion.

**Formulating a Workflow as an AOV graph.** Activity on Vertex (AOV) graph is a type of directed acyclic graph where vertices represent subtasks and edges denote precedence relations (Bondy & Murty, 2011). AOV graphs are widely used in project scheduling and management (Moder et al., 1983; Taha, 2017), helping planners visualize dependencies and sequence subtasks efficiently.

Inspired by that, we define the multi-agent workflow as an AOV graph where vertices represent subtasks, while edges denote dependencies between subtasks. Let $G = (V, E, A)$ denote the AOV graph, with $V$ the set of all subtasks (vertices), $E \subseteq V \times V$ the set of directed edges indicating subtask dependencies. For example, $e_{ij} = (v_i, v_j) \in E$ indicates that the subtask $v_i$ must be completed before the subtask $v_j$ starts. $A$ represents a set of agents for all subtasks. Each agent $a_j \in A$ is associated with a role that is responsible for executing a subset of subtasks $\mathcal{T}_j \subseteq V$.

Note that AutoGen also automatically generates subtasks and agents. However, the subtasks are designed to be executed *sequentially*. For Flow, we allow for the generation of subtasks that can run *in parallel*. This distinction enhances our framework's ability to handle multiple subtasks simultaneously, which reduces overall process time and increases efficiency.

**Modularity in a Workflow.** Modularity in system design (Baldwin & Clark, 1999) involves dividing a system into separate, independently operating modules, each responsible for specific functionalities, allowing focus on individual components without affecting the entire system. It is essential for scalability and flexibility in workflows. By reducing dependency complexity, the system can more easily adapt to changes, such as the introduction of new tasks or the reassignment of existing ones, without requiring extensive restructuring. Theorem 1 demonstrates that additional dependencies in a workflow reduce the expected success rate of subtasks. Following this conclusion, Flow advocates for the creation of subtasks that can be executed independently.

**Theorem 1.** *Consider two topologically sorted workflows A and B each consisting of N subtasks according to their execution order. Suppose*

1. *(**Random fail probability**) Each subtask $v \in \mathcal{T}$ fails with probability $p_f$, where $0 < p_f < 1$.*
2. *(**Additional dependency in Workflow B**) There exist at least one subtask $v^* \in \mathcal{T}$ and a subtask $b \in \mathcal{T}$ such that the set of immediate predecessors (dependencies) of $v^*$ in Workflow B is $D_B(v^*) = D_A(v^*) \cup \{b\}$, where $D_A(v^*)$ is the set of immediate predecessors of $v^*$ in Workflow A. For all other subtasks $v \neq v^*$ $D_A(v) \subseteq D_B(v)$.*

*The expected number of completed subtasks in Workflow A is strictly greater than in Workflow B: $E[S_A] > E[S_B]$.*

To encourage modularity in the generated AOV graph, we define two quantitative measures that evaluate ***parallelism*** and ***dependency complexity*** respectively. Parallelism measures the extent to which subtasks can be executed concurrently. Let $S_t$ represent the set of subtasks executed in the $t$ step. Let $T$ be the total number of steps (the maximum depth of the DAG). Given an AOV graph $G = (V, E, A)$, the degree of parallelism overall is defined as *the average subtask ratio over steps*:

$$P_{\text{avg}} = \frac{1}{T} \sum_{t=1}^{T} S_t.$$

Although $P_{\text{avg}}$ provides a measure of parallelism, it is insufficient to fully capture the modularity that arises when subtasks can be executed independently. Consider two workflows, both containing the same subtasks $\{A, B, C, D\}$. For Workflow 1, the task dependencies are defined as: $A \to C, B \to C, A \to D, B \to D, C \to D$. In contrast, Workflow 2 has dependencies: $A \to C, B \to C, C \to D$. Although both workflows exhibit the same level of parallelism, Workflow 2 is structurally simpler in terms of task dependencies, as it contains fewer edges.

To account for this complexity, we measure the dependency structure by analyzing the degree distribution within the subtask graph. For each subtask $v_i$, we define $\deg(v_i)$ as the number of direct connections it has on the graph $G$. The ***dependency complexity*** is quantified by the standard deviation of the number of direct connections:

$$C_{\text{dependency}} = \sigma_{\deg(v_i)} = \sqrt{\frac{1}{|V|} \sum_{v_i \in V} (\deg(v_i) - \bar{d})^2}.$$

This measure reflects the variability in the number of dependencies each subtask has, providing insight into the overall complexity of the workflow structure.

Task dependencies alone are insufficient to fully capture the modularity that allows subtasks to be executed independently. Consider Workflow 3: $A \to B \to C \to D$, which may have a similar dependency complexity to Workflow 2. However, Workflow 2 provides greater modularity and separation of subtasks, highlighting the importance of evaluating both dependency complexity and modularity to fully assess and promote effective workflow designs. Both measures are essential to ensure that subtasks can be executed in parallel while maintaining a modular approach.

---

**A Sample Prompt for Initialization $\mathcal{P}_{\text{init}}$**

```
You are an intelligent workflow planner. Given the following task requirements,
    generate a set of necessary subtasks along with their dependencies and assign
    appropriate agents to each task. Ensure that tasks that can be executed in
    parallel are identified to enhance efficiency. The workflow should be represented
    as a dictionary where each key is a task and its value contains the task's status,
    data, number of parents not completed, child tasks, and assigned agent.

Task Requirements: {TASK_REQUIREMENTS}

Output Format: { "Task_A": { "status": "not started", "data": null, "
    num_parents_not_completed": 0, "child": ["Task_B", "Task_C"], "agent": "Agent_1"
    }, "Task_B": { "status": "not started", "data": null, "num_parents_not_completed":
    1, "child": ["Task_D"], "agent": "Agent_2" }, ... }
```

**Generate an Initial AOV Graph.** Given a task requirement prompt $\mathcal{P}$, we prompt an LLM $f$ to generate a set of candidate AOV graphs $\{G_1, G_2, \ldots, G_K\}$ based on $\mathcal{P}$ and a designed ***prompt for initialization*** $\mathcal{P}_{\text{init}}$, *i.e.,* , $\{G_1, G_2, \ldots, G_K\} = f(\mathcal{P}_{\text{init}}, \mathcal{P})$. Each candidate AOV graph $G_k = (V_k, E_k, A_k)$ is *evaluated using the measures of parallelism and dependency complexity*. We prioritize the workflow with the highest parallelism score. If multiple graphs share the highest score, we select the one with the lowest dependency complexity.

Note that we prioritize parallelism and modularity early in the process and focus on refining the workflow through data-driven adjustments during running. The reasons are: 1) LLM-generated workflows possess reasoning capabilities, but may not prioritize efficiency. If parallelism and independence are not explicitly encouraged during the initial workflow generation, the applied workflow is very likely to be overly complex, which results in inefficient subtask implementation; 2) verifying correctness is inherently challenging as no additional data is available as supervised information at an early stage. As compensation, we refine the workflow by parallelism and modularity.

**Execution Plan Generation and Agent Allocation.** After we obtain the best AOV graph, a topological sort is performed on the dependency graph of the subtasks to produce a linear order of the subtasks $o : V \rightarrow \{1, 2, \ldots, |V|\}$ such that for any edge $(v_i, v_j) \in E$, $o(v_i) < o(v_j)$. The result is a sequence of subtask steps, where each step consists of subtasks that can be executed in parallel. This execution plan minimizes the number of steps needed to perform while ensuring that all subtasks are completed in the shortest possible time, adhering to their dependencies.

Each agent $a_j \in A$ is associated with a set of subtasks $\mathcal{T}_j \subseteq V$, indicating the subtasks that the agent is responsible for handling. However, if two subtasks $v_p$ and $v_q$ require the same agent $a_j$ at the same step $s_i$, we create a clone of the agent, denoted $a_j'$, to run both subtasks simultaneously without increasing the waiting time.

---

**Prompt for Update** $\mathcal{P}_{update}$

```
You are an intelligent workflow updater. Based on the current workflow and the all
    subtasks' progress data, update the workflow for acheving the objective by adding,
    removing, or modifying subtasks as necessary. Ensure that the updated workflow
    maintains modularity and maximizes parallel execution.
Output Format: { "Task_A": { "status": "not started", "data": null,  ... }
```

---

**Workflow Refinement and Dynamic Updating.** We leverage LLM as a global inspector to continuously monitor task progress and dynamically modify the AOV graph based on global information when necessary. Specifically, given the task requirements prompt $\mathcal{P}$ and the update prompt $\mathcal{P}_{\text{update}}$, the current AOV graph $G^t$, and the generated data $D^t$ containing the status of subtasks and the output of agents to run subtasks. Similarly to the initialization process, we generate $K$ candidate graphs: $\{G_1^{t+1}, G_2^{t+1}, \ldots, G_K^{t+1}\} = f(\mathcal{P}_{\text{update}}, \mathcal{P}, D^t)$. We follow the same selection strategy as in initialization, which prioritizes the workflow with the highest parallelism score and further selects the one with the lowest dependency complexity if multiple graphs share the highest parallelism score.

With the modularity constraint introduced in previous sessions, our dynamic updates can largely fulfill flexibility, allowing modifications to subtask allocations including *deletion*, *addition*, *editing*, *rerunning*, and *reassignment* of agents without necessarily affecting other agents or their assigned subtasks. Namely, one subtask $v_i$ can be replaced by a collection of subtasks $V$ after updating. This unique advantage is particularly beneficial when subtask requirements become more challenging, as subtask dependencies can be highly complex.

Note that with sufficient data and computational resources, we could further enhance our framework by fine-tuning LLM with reinforcement learning for workflow generation. For example, the LLM would be trained to maximize a reward function designed around key performance indicators such as task completion speed, resource utilization, and minimization of workflow disruptions.

**Implementation.** Our framework employs a dictionary-based structure, $\tilde{G}$, to efficiently manage and dynamically update workflows within a multi-agent framework. Each subtask $v$ in the workflow is represented as a key in $\tilde{G}$, the value being another dictionary that encapsulates various attributes of the subtask. The structure is specifically defined as:

$$\tilde{G}[v] = \{\text{"subtask requirement"}, \text{"status"}, \text{"data"}, \text{"num\_parents\_not\_completed"}, \text{"child"}, \text{"agent"}\}.$$

In each $\tilde{G}[v]$, the values of each key are as follows:

- "subtask requirement": the text of the task requirement;
- "status": the current task implementation status *e.g.* "not started", "in progress", "completed";
- "data": data relevant to this task;
- "num_parents_not_completed": the count of uncompleted parent tasks to manage dependencies;
- "child": a list of child tasks that depend on the current task's completion;
- "agent": the agent assigned to the task.

This dictionary-based structure can be converted directly to JSON, and the organized information is easily readable and summarizable by LLM, granting our system inherent simplicity and flexibility. In addition, each subtask execution readiness is determined by the attribute "num_parents_not_completed". Subtasks with a count of zero are eligible to run concurrently, leveraging our system's capability to handle parallel subtask execution effectively. Upon completion of each subtask, we perform a systematic review to determine if the workflow requires refinement, ensuring that all dependencies are accurately accounted for and that the workflow remains aligned with project goals. In addition to monitoring the subtask completion by the "status" and "num_parents_not_completed" counts reported by agents. Flow also double-checks the completion of each subtask by asking if all the requirements of this subtask are fulfilled. This will largely prevent errors from inaccurate reporting by agents or unforeseen system anomalies. This rigorous verification process enhances the reliability and integrity of our workflow management system.

## 4 EXPERIMENTS

**Baselines.** In all experiments, we compare Flow to existing multi-agent frameworks: (1) AutoGen , (2) Camel , and (3) MetaGPT . In our experiments, we use agents empowered by GPT-4o-mini and GPT-3.5-Turbo (OpenAI, 2024).

**Experiment Design.** We designed three diverse and engaging tasks to evaluate multi-agent collaboration frameworks: 1) website design, 2) LaTeX Beamer writing, and 3) gobang game development. The rationale for selecting coding-based experiments is twofold. First, most multi-agent frameworks, such as MetaGPT , are optimized for coding and writing tasks. Using non-coding tasks could introduce bias. Second, coding tasks effectively showcase the ability of a framework to assign agents and manage task allocation.

*Gobang Game Development*: This task requires creating a gobang game with a user interface and a simple AI opponent. Players can choose between black or white stones, with the UI clearly indicating turns and announcing the winner or draw when the game ends. This task demonstrates the framework's ability to handle modular design and task parallelism, as it involves coordinating game logic, AI implementation, and user interface development simultaneously.

*LaTeX Beamer Writing*: This task focuses on generating LaTeX slides that cover reinforcement learning algorithms, including motivations, problem statements, intuitive solutions, and detailed mathematical equations. A specific page requirement is to test the framework's ability to follow instructions precisely. The task highlights the framework's parallel processing capabilities of simultaneous generation of content, formatting, and presentation structure. The structured format of LaTeX also tests how effectively the framework manages modularity and concurrent tasks.

*Website Design*: This task builds a professional website for the International Conference on Learning Representations, hypothetically scheduled in San Francisco from April 27 to May 1, 2025. The website must feature key elements such as a detailed conference schedule and venue information with an interactive map. This task assesses frameworks' ability to manage parallel workflows and modular components, including user interface design, functionality, and adherence to design guidelines, showcasing how well a framework can handle complex task decomposition and execution.

### 4.1 EVALUATIONS OVER THREE DESIGNED TASKS

**Evaluation Metrics.** To conduct both quantitative and qualitative evaluations, we employ two metrics: ***Success Rate*** and ***Human Rating***. The success rate is a quantitative measure that ranges from 0 to 1. It assesses whether the multi-agent framework successfully generates executable outputs

Table 1: Comparison of different multi-agent frameworks on Gobang Game Development

| Model | Success Rate (%) | | | | Human Rating |
|---|---|---|---|---|---|
| | Compilable | Interactable | Game Rule | Overall Score | (1-4) |
| AutoGen | 80 | 60 | 40 | 60 | 2.26 |
| MetaGPT | **100** | **100** | 20 | 73 | 1.24 |
| CAMEL | 40 | 40 | 0 | 27 | 2.50 |
| **Flow (Ours)** | **100** | **100** | **100** | **100** | **4.00** |

Table 2: Comparison of different multi-agent frameworks on LaTeX Beamer Writing

| Model | Success Rate (%) | | | | Human Rating |
|---|---|---|---|---|---|
| | Compilable | Completeness | Page Limit | Overall Score | (1-4) |
| AutoGen | 80 | 80 | 40 | 67 | 3.00 |
| MetaGPT | 80 | 80 | 20 | 60 | 1.83 |
| CAMEL | **100** | **100** | 0 | 66 | 1.83 |
| **Flow (Ours)** | **100** | **100** | **100** | **100** | **3.33** |

Table 3: Comparison of different multi-agent frameworks on Website Design

| Model | Success Rate (%) | | | | Human Rating |
|---|---|---|---|---|---|
| | Compilable | Basic Information | Sections | Overall Score | (1-4) |
| AutoGen | 80 | 80 | 60 | 73 | 2.62 |
| MetaGPT | **100** | **100** | 40 | **80** | 1.72 |
| CAMEL | 80 | 80 | 0 | 53 | 2.02 |
| **Flow (Ours)** | 80 | 80 | **80** | **80** | **3.28** |

that fully meet the task requirements. A higher score indicates a greater level of success in accurately fulfilling the task objectives. Different tasks may have different evaluation metrics. The description for each evaluation metric is defined in Appendix B.1, B.2 and B.3. Human ratings are used to evaluate the quality of the generated results in alignment with the task description. We gathered 50 participants with programming and machine learning backgrounds to rank the outcomes produced by different methods. A detailed description of how we take scores is shown in Appendix A.

**Summary.** We summarize the performance of different methods on three tasks from Table 1, 2 and 3, comparing the overall score with respect to the success rate and human rating. For Flow, the overall score and human rating over three tasks are (100, 4) on game development, (100, 3.33) on LaTeX writing, and (80, 3.28) on website design. Thus, the average performance of Flow is a 93% success rate and 3.54 out of 4 in human satisfaction. Similarly, we have the average performance of AutoGen as (66.7, 2.63), MetaGPT as (71, 1.60), and CAMEL as (48.67, 2.12). Overall, our method Flow has completed tasks with the most satisfaction and the highest success rate. Information about Flow's workflow on those tasks is in Appendix D.

## 4.2 RESULT FOR GOBANG GAME DEVELOPMENT

The experimental setup is thoroughly detailed in Appendix B.2 and the visualization result is shown in Fig. 1. As shown in Table 1, Flow achieves a 100% success rate across all aspects, along with the highest human satisfaction. More explanations for each framework are given below.

*AutoGen*: AutoGen was evaluated across five trials. Of these, four produced valid, executable code. One instance encountered a runtime error that impacted execution, and another revealed a minor game interface bug. However, in the remaining two trials, the game flow proceeded correctly, though the chess pieces appeared as the text "black" and "white" instead of graphical icons.

*MetaGPT*: MetaGPT produced executable code in all five attempts, demonstrating its overall capability for code generation. However, in four of these runs, it unexpectedly created a Tic-Tac-Toe game rather than Gobang. Only one of these trials delivered a fully functional version that allowed both the user and the AI to take turns properly, concluding the game once the winning condition was met.

*CAMEL*: CAMEL was tested five times, successfully generating executable code in two cases. However, these successes did not include an AI opponent. The remaining three trials were non-

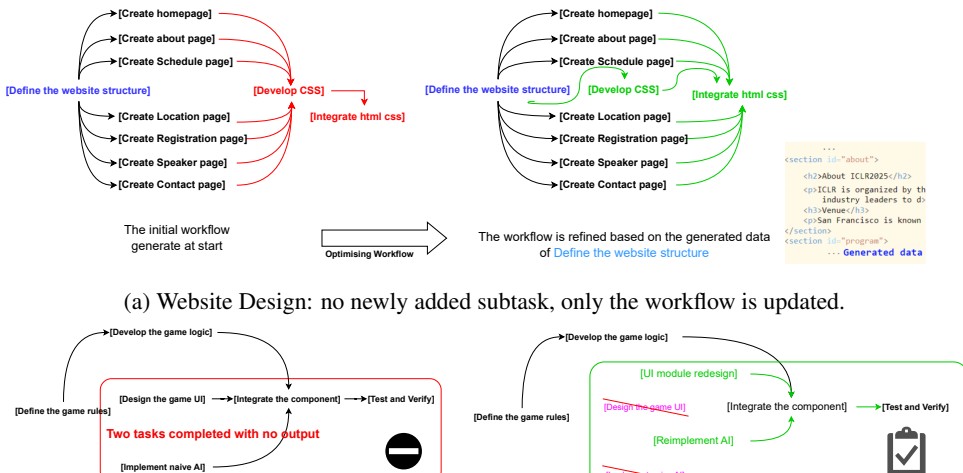

(a) Website Design: no newly added subtask, only the workflow is updated.

(b) Gobang Game Development: adding two new subtasks to replace bad ones.

Figure 3: Workflow and dynamic update in two experiments.

executable due to various errors. Despite these hurdles, CAMEL's success cases demonstrate its basic proficiency in generating operational game logic.

*Flow*: Under the same test conditions, Flow consistently produced valid outputs in all five attempts without any observable errors. The game behaved as intended, enabling both the player and a straightforward AI to alternate moves. Flow also successfully rendered the board using black and white piece icons, offering a more visually engaging interface compared to text-only labels.

### 4.3 RESULT FOR LaTeX BEAMER WRITING

Experimental results are presented in Table 2 with the following explanations:

*AutoGen*: Across five trials, AutoGen successfully generated an output in all five trials. However, in one case, the LaTeX file failed to compile due to syntax errors, and in two instances, the outputs did not meet the required length. The remaining trials produced documents that satisfied both length and content requirements.

*MetaGPT*: We ran MetaGPT five times, and in four of these trials, it produced valid LaTeX files. The primary issue in the remaining trial was that Python code was included within the ".tex" file. Although the documents met the required content specifications, only one met the prescribed page length of either 30 or 20 pages.

*CAMEL*: CAMEL generated five valid ".tex" files that successfully compiled into Beamer presentations, each containing the necessary sections such as motivation. However, none achieved the specified page length of 30 or 20 pages in our tests.

*Flow*: Flow consistently produced LaTeX compilable outputs across five trials. While one output included some repetitive content, the resulting Beamer presentations met the specified length requirements and covered all required material.

### 4.4 RESULT FOR WEBSITE DESIGN

Similarly to the previous two, the detailed experiment setup is in Appendix B.3. Here we illustrate the results in Table 3 as follows:

*AutoGen*: AutoGen successfully produced HTML websites in four out of five trials, showcasing a generally reliable code-generation process. However, in one trial, the output contained only minimal

content, just a few sentences per section—indicating challenges in generating more detailed or feature-rich outputs.

*MetaGPT*: MetaGPT generated valid HTML and CSS in all five attempts, showcasing its ability to produce functional web content. However, the outputs were relatively simple, lacking advanced design elements, and often omitted essential sections such as a dedicated venue section and a map.

*CAMEL*: CAMEL successfully delivered four out of five executable websites. However, when additional complexity was required, the results were occasionally incomplete. In one instance, it produced only the HTML file, omitting the accompanying CSS, indicating potential challenges in managing multi-file dependencies.

*Flow*: Flow successfully generated fully functional websites in four out of five trials. In these cases, the websites featured detailed descriptions and essential interactive elements. However, in one trial, errors occurred when integrating different files, preventing successful execution.

## 5 WORKFLOW UPDATE

**Update based on Generated Data.** Fig. 3(a) demonstrates the update process of Flow in the website design example. Upon completion of the first subtask, the system identifies potential changes and redundancies, triggering a restructuring process to improve efficiency. Once the subtask "Define the website structure" is completed, the generated data, which includes HTML structures and elements, is sufficient to proceed with the CSS creation. As a result, the workflow is updated to incorporate the development of CSS based on the completed "Define the website structure" subtask.

Fig. 3(b) illustrates a result of our dynamic updating process, where the system, upon receiving information about completed subtasks, decides to add a bridging subtask to handle gaps and ensure that the workflow continues smoothly.

**Error Handling.** To evaluate the effectiveness of our update mechanism, we intentionally introduced random masking to certain subtask outputs, replacing them with "none" before passing them to the next agent. We conducted five trials and recorded the success scores. Since other frameworks employ a sequential workflow, we limit the comparison to our own approach in this context.

Table 4: Success Rate (%) of Error handling with dynamically updating.

| Task | Flow w/o Update | Flow |
| --- | --- | --- |
| Website Design | 46 | **87** |
| Gobang Game Development | 0 | **93** |
| LaTeX Beamer Writing | 67 | **93** |

We observed a significant difference in the success rate between using dynamic update and not, particularly in the Interactive Game section as shown in Table 4. The main issue arises when the previous agent fails to provide the necessary information, yet the second agent continues with its subtask, leading to a major disconnect in the code. This often results in Python being unable to compile due to missing or mismatched components. Similarly, in website design, the lack of required elements caused by this failure impacts the overall functionality and structure. During the execution of subtasks, errors may arise due to the limitations of the LLM-based agent or underperformance in certain tasks. Therefore, the ability to dynamically update the agent workflow to address such issues is essential.

## 6 CONCLUSION

We present Flow, a novel LLM-based multi-agent framework that can dynamically adapt to unforeseen challenges for general task executions. By dynamically updating the agentic workflow using AOV graphs, our framework has largely fulfilled the modularity requirements to complete complex tasks. We demonstrate our method through case studies on a series of experiments, ranging from website design, game development, and LaTeX Beamer writing, Through objective evaluation metrics and human feedback, we found that Flow improves execution efficiency, offers better error tolerance, and delivers overall stronger performance.

## ETHIC STATEMENT

Our work follows the ICLR Code of Ethics. All data used are anonymized, eliminating any potential privacy concerns. The study was conducted in compliance with the ethical guidelines of The University of Adelaide Human Research Ethics Committee with approval number H-2025-085.

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

# APPENDIX

## CONTENTS

## A  HUMAN EVALUATION PROCESS

Sometimes, LLM can correctly fulfill each requirement of a task, but the quality of completion may vary. In such cases, human evaluation is necessary to assess the quality of the output. For each task, the final output of each multi-agent framework was evaluated by 50 participants, who ranked the outputs from best to worst. Points were awarded based on the rankings, with the 1st place receiving 4 points, the 2nd place receiving 3 points, etc. The final result was determined by calculating the average score. The detailed distribution is shown in Fig. 5.

## B  EXPERIMENT SETUPS

### B.1  EXPERIMENT SETUP: LATEX BEAMER WRITING

---

**User input**

```
I am a lecturer teaching a machine learning course to research students, I am
    preparing lecture slides on various reinforcement learning algorithms.
Note that:
1). Given that the lecture duration is 2 hours, the slides should span approximately
    30 pages.
2). For each reinforcement learning algorithm covered, the slides will include the
    following key components: the motivation behind the algorithm, the problem it aims
     to solve, an intuitive solution, and the detailed mathematical equations that
    underpin the method.
3).  It is essential that the lecture is comprehensive and self-contained, providing
    students with a clear understanding of each algorithm from both a conceptual and
    technical perspective.
```

---

The task involves generating a LaTeX Beamer presentation, which is a popular LaTeX class used to create professional-quality slides with various templates and effects. In this experiment, the objective is to produce presentations with different configurations, assessing the framework's ability to follow instructions. The experiment includes the following configurations:

Config 1:  A 30-slide presentation, including motivation, problem statement, intuitive solution, and detailed mathematical equations.

Config 2:  A 20-slide presentation, including motivation, problem statement, intuitive solution, and detailed mathematical equations.

Config 3:  A 30-slide presentation, including motivation, problem statement, intuitive solution, and pseudocode.

Config 4:  A 20-slide presentation, including only motivation and intuitive solution.

Config 5:  A 30-slide presentation, including motivation, problem statement, intuitive solution, and detailed mathematical equations.

The goal is to examine the framework's ability to follow specific instructions while generating over 20 and 30 slides in different scenarios.

This task is well-suited for evaluation because it requires not only text generation but also an understanding of formatting and presentation logic. It serves as a comprehensive test of multitasking and reasoning capabilities. The structured nature of LaTeX allows for a rigorous assessment of the agent's ability to manage complex, multicomponent tasks.

**Evaluation Metrics:** The following metrics are used to assess the performance of the generated LaTeX Beamer writing:

(1) **Compilable:** Verifies whether the generated LaTeX code can compiles into a valid Beamer presentation or not. A successful compilation is rewarded with a score of 1, otherwise 0.

(2) **Completeness:** Ensures that the final Beamer presentation includes all required components like: motivation, problem, intuitive solution, and equations. Missing any of these results in a score of 0.

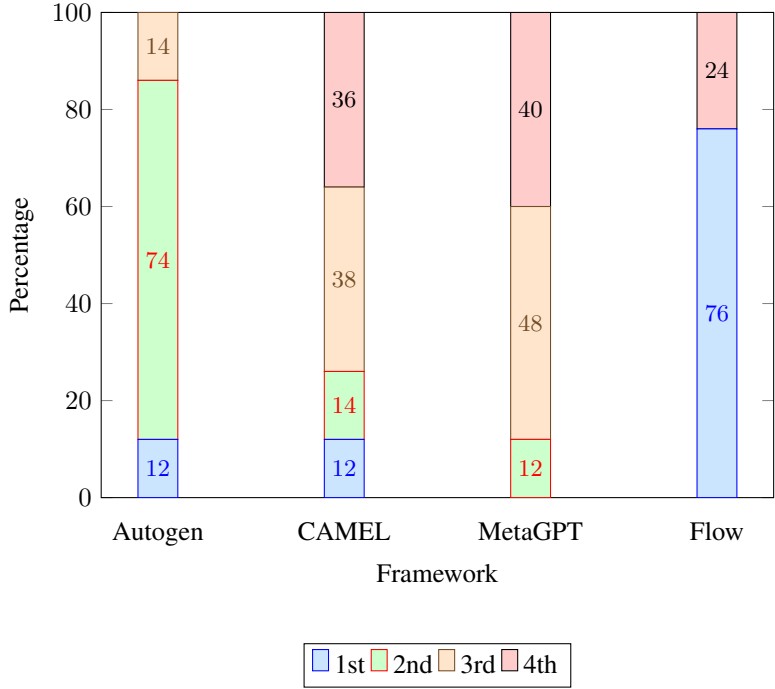

Figure 4: Ranking distribution for website design across different frameworks. The results indicate that our method (Flow) outperforms others by achieving the highest percentage of first-place rankings.

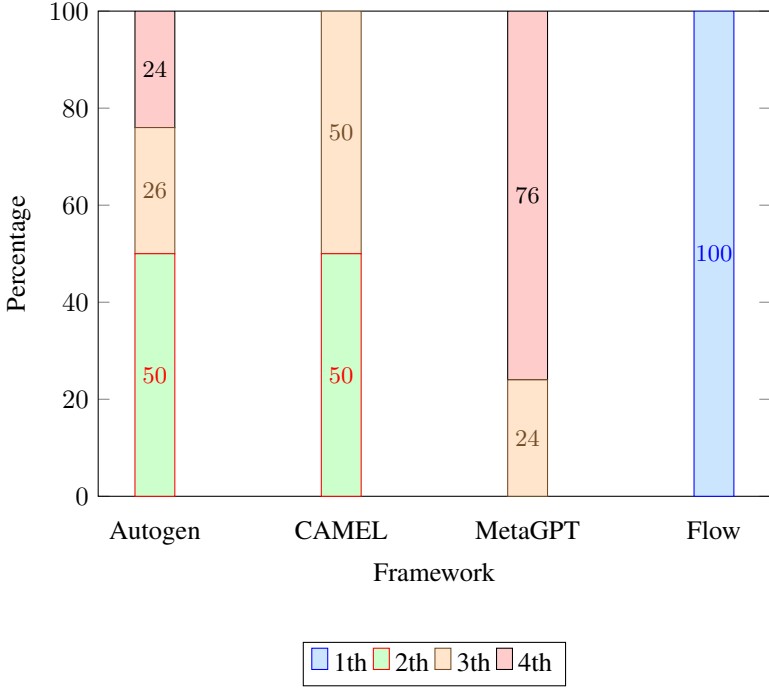

Figure 5: Ranking distribution for gobang game development across different frameworks. The results indicate that our method (Flow) outperforms others by achieving the highest percentage of first-place rankings.

(3) **Page Limit:** Assesses whether the presentation adheres to the specified page limits as outlined in the prompt.

The final result is calculated as the average of these three scores and is shown as a percentage.

## B.2 EXPERIMENT SETUP: GOBANG GAME DEVELOPMENT

> **User input**
>
> ```
> I am developing a Gobang game that includes a naive AI and a user interface. The game
>     should end when either a player wins or the board is completely filled. The user
>     interface must clearly indicate whose turn it is and display a message when the
>     game concludes, specifying the winner. Additionally, the user should have the
>     option to play as either black or white stones.
> ```

Gobang, also called "Five in a Row", is a strategy board game where two players take turns placing black and white pieces on a grid. The objective is to be the first to align five consecutive pieces in a horizontal, vertical, or diagonal line. This experiment assesses our framework's ability to efficiently develop the game by utilizing parallelism to divide the development process into smaller, manageable tasks, such as game logic, AI move generation, and user interface (UI) design. We apply the same approach, taking the average score from five trials.

**Evaluation Metrics:** The following metrics are used to assess the performance of the generated Gobang game:

(1) **Compilable:** The code compiles without errors. Any error that causes termination will result in a score of 0.

(2) **Interactable:** Properly supports both user and AI movements. If both functions are achieved, score 1 else 0.

(3) **Game Rule:** Ends correctly when five pieces are aligned, correct terminated will result in 1 final score.

Each of these metrics is scored as 0 or 1, and the final result is calculated as the average of these scores and turned into a percentage. These metrics allow for a comprehensive assessment of the efficiency, accuracy, and adaptability of each framework in developing a functional Gobang game with AI capabilities.

## B.3 EXPERIMENT SETUP: WEBSITE DESIGN

> **User input**
>
> ```
> I am designing a website for the International Conference on Learning Representations
>     (ICLR2025), which will take place from April 27, 2025, to May 1, 2025, in San
>     Francisco, California, United States. The conference is organized by the
>     International Association for Learning Representations.
> Note that:
> 1). For each section, I would like to see example HTML content. Additionally, a sample
>      CSS stylesheet should be provided to style the website. The content must be
>     professional, clear, and appropriate for an international academic conference.
> 2). The website should include all the provided details, including a comprehensive
>     conference schedule and a section dedicated to the conference venue, featuring a
>     map.
> ```

We tasked the frameworks with developing a comprehensive website for the ICLR conference to evaluate their ability to handle complex tasks that require both flexible task coordination and effective problem solving. This task tested the ability of the frameworks to manage multiple interdependent steps, such as designing user interfaces, ensuring functionality, and adhering to specific design guidelines.

**Evaluation Metrics:** The following metrics are used to assess the performance of the generated website:

(1) **Compilable:** Checks if the HTML renders into a functioning website, If yes then score 1, can't render will result of score 0

(2) **Basic Information:** Verifies the presence of essential details like conference name, date, location, and organizer. Missing any of the information will caused the score to be 0

(3) **Sections:** Ensures inclusion of all required sections, with a focus on the schedule and venue as prompt asked. Missing the required part in the prompt will result in a score of 0 in score.

By presenting a real-world scenario involving intricate requirements, we were able to observe how well the frameworks could break down a large project into manageable components and coordinate efforts across different tasks.

## B.4 HOW DIFFERENT LLM AFFECT UPDATES

To verify how our framework performs with different capabilities of LLM, we test both GPT-4o-mini and GPT-3.5-Turbo on three tasks we designed. In this experiment, each task was run five times on different models, and the average of the results was calculated as the final outcome. We recorded three metrics: average init task, average changed task, and average changed ratio.
**Init task** refers to the number of subtasks that need to be executed within the workflow after selecting the optimal workflow but before the execution begins.
**Average changed task** indicates the number of subtasks in the original workflow that were updated after the execution of the workflow.
**Average changed ratio** is calculated by dividing the average changed task by the init task, providing a more intuitive reflection of the proportion of subtasks that were updated.

Table 5: Update information on GPT-3.5-Turbo and GPT-4o-mini

| LLM-Agent | Task | Initial Tasks (avg.) | Change Ratio (avg.) |
|---|---|---|---|
| GPT-3.5-Turbo | Gobang Game Development | 7.8 | 44 |
| | Website Design | 7.2 | 66 |
| | LaTeX Beamer Writing | 4.4 | 71 |
| GPT-4o-mini | Gobang Game Development | 8 | 35 |
| | Website Design | 7.2 | 47 |
| | LaTeX Beamer Writing | 9.2 | 53 |

## B.5 HOW DIFFERENT LLM AFFECT PERFORMANCE

In this experiment, we used the GPT-3.5-Turbo model to conduct experiments on three tasks in different frameworks. Each task was executed five times. We evaluated the results using the same scoring metrics described above.

Table 6: Comparison of LLM-based multi-agent frameworks on Gobang Game Development with GPT-3.5-Turbo

| Model | Success Rate (%) | | | |
|---|---|---|---|---|
| | Compilable | Interactable | Game Rule | Overall Score |
| AutoGen | 80 | 20 | 20 | 40 |
| MetaGPT | 80 | 20 | 40 | 53 |
| CAMEL | 80 | 80 | 40 | 67 |
| **Flow (Ours)** | **100** | **100** | **60** | **87** |

Table 7: Comparison of LLM-based multi-agent frameworks on Website Design with GPT-3.5-Turbo

| Model | Success Rate (%) | | | |
|---|---|---|---|---|
| | Compilable | Basic Information | Sections | Overall Score |
| AutoGen | 20 | 0 | 0 | 7 |
| MetaGPT | 80 | 60 | **60** | 67 |
| CAMEL | 40 | 40 | 20 | 33 |
| **Flow (Ours)** | **100** | **100** | 40 | **80** |

Table 8: Comparison of LLM-based multi-agent frameworks on LaTeX Beamer Writing with GPT-3.5-Turbo

| Model | Success Rate (%) | | | |
|---|---|---|---|---|
| | Compilable | Completeness | Page Limit | Overall Score |
| AutoGen | 40 | 0 | 0 | 13 |
| MetaGPT | 20 | 20 | 0 | 13 |
| CAMEL | 80 | 80 | 0 | 53 |
| **Flow (Ours)** | **100** | **100** | 0 | **67** |

Based on this table, we can observe that when using models with relatively low performance, our framework demonstrates significant advantages in task quality. Overall, even when using less powerful LLM like GPT-3.5-Turbo, our framework consistently maintains a high standard of performance.

## B.6 TIME COST OF DIFFERENT BASELINE

To quantitatively measure the cost of our framework, we use execution time as a standard. Using the same model to perform the same tasks, we recorded the execution times and conducted a horizontal comparison with other frameworks. Each task was executed five times, and the average execution time was calculated.

| Task | Flow (w/o update) | Flow (w/ update) | MetaGPT | CAMEL | AutoGen |
|---|---|---|---|---|---|
| **GPT-3.5-Turbo** | | | | | |
| Gobang Game | 26.12 ± 11.35 | 33.57 ± 12.46 | 34.00 ± 15.12 | 121.52 ± 20.87 | 31.00 ± 14.67 |
| Website Website | 23.46 ± 10.84 | 34.23 ± 13.12 | 85.14 ± 18.52 | 41.96 ± 12.89 | 44.00 ± 15.34 |
| Latex Beamer | 18.34 ± 9.73 | 24.12 ± 10.89 | 29.92 ± 14.87 | 166.00 ± 22.64 | 31.00 ± 16.78 |
| **GPT-4o-mini** | | | | | |
| Gobang Game | 60.45 ± 14.78 | 72.34 ± 13.45 | 99.45 ± 16.92 | 110.94 ± 19.67 | 148.72 ± 25.34 |
| Website Website | 51.98 ± 20.19 | 52.14 ± 14.89 | 127.49 ± 17.52 | 74.53 ± 18.34 | 86.78 ± 21.23 |
| Latex Beamer | 53.19 ± 17.65 | 83.34 ± 15.89 | 66.72 ± 19.45 | 106.34 ± 20.78 | 95.21 ± 22.56 |

Table 9: Comparison of task performance across different framework, including standard deviations. The standard deviations reflect realistic variability with increased variance across tasks and framework.

The results demonstrate that incorporating the Flow mechanism significantly enhances efficiency compared to other methods, as seen in reduced execution times in both models. However, the introduction of updates incurs additional computational overhead, resulting in a noticeable increase in execution time, highlighting the trade-off between adaptability and efficiency. Nonetheless, Flow maintains faster execution times compared to several other frameworks.

## C    CUSTOM METRICS FOR PARALLELISM AND DEPENDENCY

### C.1    PARALLELISM METRICS

Speedup ($S = \frac{T_1}{T_p}$), this metric measures the ratio of execution time on a single processor ($T_1$) to that on multiple processors ($T_p$). While effective in frameworks where these times can be measured, it requires actual execution on both single and multiple processors. In our case, such execution times are not readily obtainable because our focus is on task-solving workflows rather than on processing workloads that can be easily benchmarked in this way.

Amdahl's Law ($S(p) = \frac{1}{f_s + \frac{1-f_s}{p}}$) and Gustafson's Law ($S(p) = p - f_s \cdot (p - 1)$), both laws require knowledge of $f_s$, the proportion of the task that is inherently serial, and $p$, the number of processors. Our task graphs have complex dependency structures, where tasks cannot be neatly categorized as strictly "serial" or "parallel." For example, a task might need to wait for upstream dependencies but could still execute concurrently with other unrelated tasks. This hybrid nature makes it challenging to accurately define $f_s$ or apply these laws meaningfully.

### C.2    DEPENDENCY METRICS

Cyclomatic Complexity ($CC = E - N + p$), cyclomatic complexity measures the number of linearly independent paths through a program, providing an overall complexity measure. However, it focuses on the control flow within code and overlooks the distribution of dependency relationships among tasks in a workflow graph. It does not capture the "dependency concentration" or "dispersion," which are crucial to understanding the impact of dependencies on workflow robustness and the ease with which LLM can comprehend and update the workflow.

### C.3    PROPOSED METRICS FOR TASK WORKFLOW EVALUATION

Given these limitations, we use two simple metrics in our LLM-based multi-agent framework:

1). Parallelism Metric: This metric does not rely on execution time measurements or require assumptions about tasks being strictly serial or parallel. It directly reflects the workflow's potential for concurrent task execution, making it more applicable to our scenario.

2). Dependency Metric: We focus on the "dependency concentration" or "dependency dispersion" by analyzing the standard deviation of the degree distribution in the task graph. This metric provides an intuitive reflection of critical dependency points within the workflow. By highlighting how dependencies are distributed among tasks, it helps us understand and mitigate potential bottlenecks, enhancing both robustness and the LLM's ability to process workflow updates efficiently.

## D    EXAMPLES OF FLOW'S WORKFLOW

In this section, we present examples of actual workflows generated by Flow.

Fig. 6 showing Flow's workflow in generating LaTeX Beamer, Flow concurrently generates the four required components for each algorithm: motivation, problem, intuitive solution, and mathematical equations.

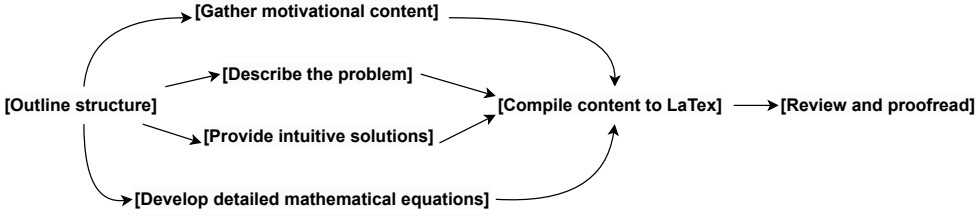

Figure 6: Workflow of LaTeX Beamer Writing in Flow

For the task of developing a gobang game, Flow recognizes that the UI and main game logic can be separated and executed in parallel to enhance overall speed and efficiency, as shown in Fig. 7. Additionally, there remains a clear sequential process; for instance, the game rules must be defined first before the corresponding code can be deployed.

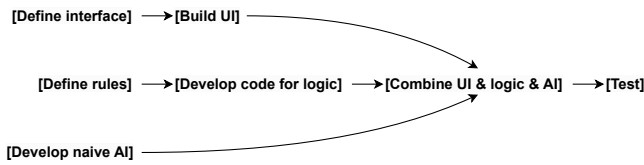

Figure 7: Workflow of Gobang Game Development

For the task of website design, as shown in Fig. 8, Flow treats different parts of the HTML as individual subtasks, which helps to increase overall speed. Additionally, dividing the process into separate components allows for parallel execution and improved modularity, ensuring that if an issue arises in one part of the HTML, it will not impact the performance of other sections. This approach improves both efficiency and fault tolerance.

Figure 8: Workflow of Website Design

## D.1 EXAMPLE WORKFLOW

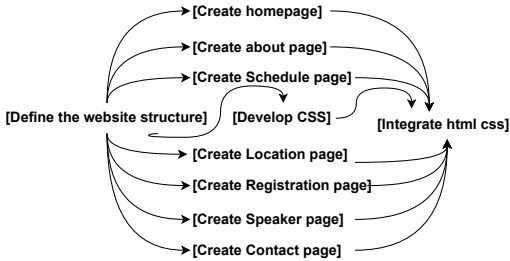

Figure 9: A workflow of Website Design in VSCode

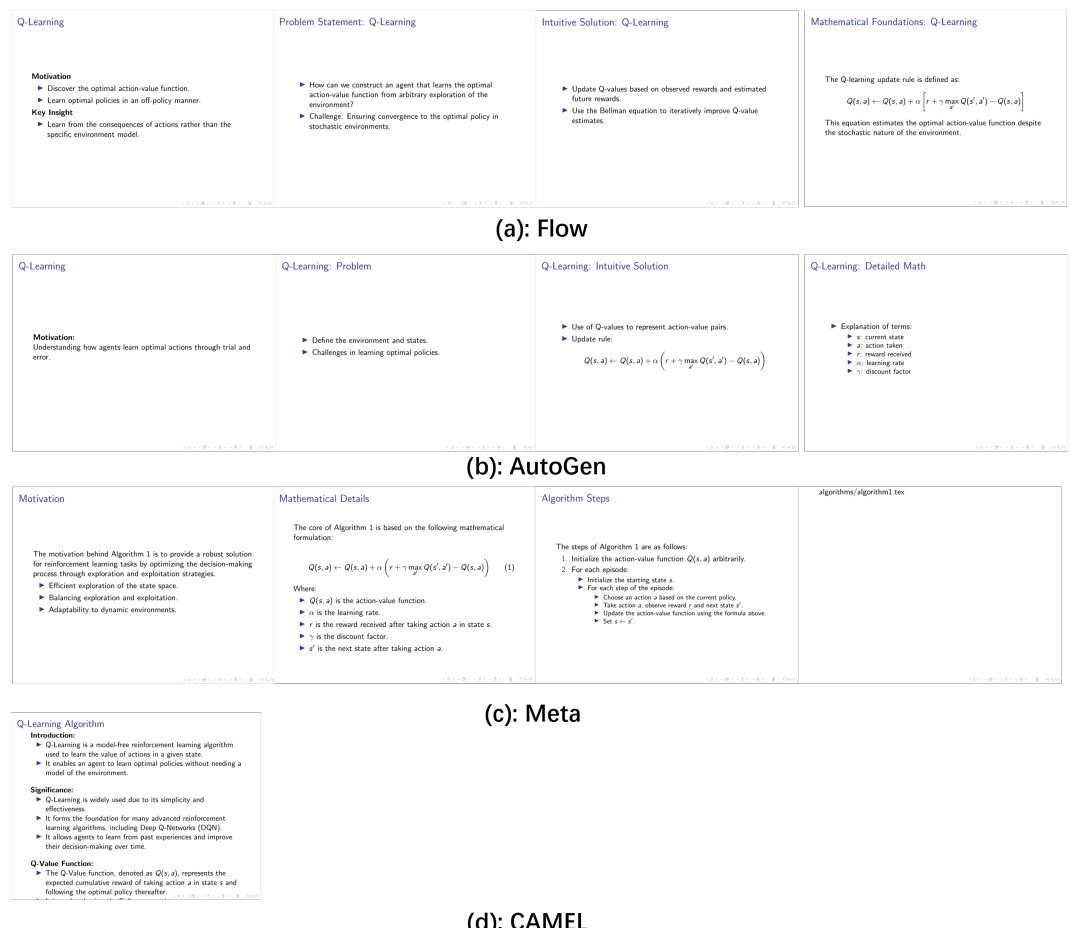

Figure 10: Different multi-agent frameworks' LaTeX Beamer

## D.2 Pseudocode for updating AOV

**Algorithm 1:** Helper Function for Updating Graph

---

1 **Function** `UpdateGraph(`$\tilde{G}, \mathcal{P}, \mathcal{P}_{init}$`)`**:**
       // Generate updated candidate workflows using LLM
2     $\{\tilde{G}_1, \tilde{G}_2, \ldots, \tilde{G}_K\} \leftarrow f(\tilde{G}, \mathcal{P}, \mathcal{P}_{init})$;
       // Initialize selection variables
3     $P_{\max} \leftarrow -\infty$;
4     $C_{\min} \leftarrow +\infty$;
5     $\tilde{G}_{\text{optimal}} \leftarrow \text{None}$;
       // Evaluate each candidate workflow
6     **for** *each candidate workflow* $\tilde{G}_k$ *in* $\{\tilde{G}_1, \tilde{G}_2, \ldots, \tilde{G}_K\}$ **do**
7         Compute Parallelism $P_k \leftarrow P_{\text{avg}}(\tilde{G}_k)$;
8         Compute Dependency Complexity $C_k \leftarrow C_{\text{dependency}}(\tilde{G}_k)$;
9         **if** $P_k > P_{max}$ *or* $(P_k == P_{max}$ *and* $C_k < C_{min})$ **then**
10             $P_{\max} \leftarrow P_k$;
11             $C_{\min} \leftarrow C_k$;
12             $\tilde{G}_{\text{optimal}} \leftarrow \tilde{G}_k$;

13     **return** $\tilde{G}_{optimal}$;

---

---

**Algorithm 2:** Flow

---

**Data:** Task Requirements $\mathcal{P}$, Initialization Prompt $\mathcal{P}_{\text{init}}$, Update Prompt $\mathcal{P}_{\text{update}}$
**Result:** Optimized Multi-Agent Workflow

```
// Step 1:  Implement a Workflow using a dictionary structure
```
1 Initialize workflow formulation by defining the task dictionary $\tilde{G}$ where each key $v \in V$ maps to a dictionary containing: $\tilde{G}[v] = \{\text{status}, \text{data}, \text{num\_parents\_not\_completed}, \text{child}, \text{agent}\}$

```
// Step 2:  Generate an Initial Workflow
```
2 $\tilde{G} \leftarrow \text{UpdateGraph}(\{\}, \mathcal{P}_{\text{init}}, \mathcal{P})$;

```
// Step 3:  Workflow Refinement and Dynamic Updating
```
3 **while** *there exists at least one sub-task in $\tilde{G}$ that is not completed* **do**
4    **if** *an update to the workflow is required* **then**
```
        // Generate and Select the Best Updated Workflow
```
5       $\tilde{G} \leftarrow \text{UpdateGraph}(\tilde{G}, \mathcal{P}_{\text{update}}, \mathcal{P})$;
6       Update workflow dictionary $\tilde{G}$ to $\tilde{G}_{\text{best}}$;
```
        // Regenerate Execution Plan and Reallocate Agents
```
7       Perform Topological Sort on $\tilde{G}$ to obtain updated execution order $\sigma$;
8       Assign agents $A_j$ to their respective sub-tasks $\mathcal{T}_j \subseteq V$;
9    **end**
```
    // Execute Available Sub-tasks in Parallel
```
10    **foreach** *sub-task $v_i \in V$* **do**
11       **if** *status of $v_i$ is **not started** and $\tilde{G}[v_i].num\_parents\_not\_completed == 0$* **then**
12          **if** *agent $a_j$ is available* **then**
13             Assign agent $a_j$ to sub-task $v_i$;
14          **else**
15             Clone agent $a'_j$;
16             Assign cloned agent $a'_j$ to sub-task $v_i$;
17          **end**
```
            // Execute subtask v_i in parallel
```
18          Execute $v_i$ using agent $a_j$ or cloned agent $a'_j$ concurrently;
```
            // Update Subtask Status and Data
```
19          Update status of sub-task $v_i$ to **in progress**;
```
            // After execution, update related data
```
20          Update output of subtask $v_i$ to $\tilde{G}[v_i].data$;
21          $\tilde{G}[v_i].status \leftarrow$ "completed";
```
            // Update Child Tasks' Parent Completion Count
```
22          **foreach** *child task $c \in \tilde{G}[v_i].child$* **do**
23             $\tilde{G}[c].num\_parents\_not\_completed \leftarrow \tilde{G}[c].num\_parents\_not\_completed - 1$;
24          **end**
25       **end**
26    **end**
27 **end**

---

### D.3 PROMPT FOR WORKFLOW UPDATE

```
User input

1. Update the Workflow

    - Evaluate Completed Tasks:
        - Focus: Examine only tasks with '"status": "completed"'.
        - Check Data:
            - Ensure that '"data"' for each task is sufficient, detailed, and directly
                contributes to the 'final_goal'.

    - Assess Workflow Structure:
        - Examine All Tasks: Review all tasks, including those labeled '"completed"',
            '"pending"', and '"in-progress"'.
        - Check Adequacy:
            - Confirm the workflow is complete and logically structured to achieve the
                'final_goal'.
            - Ensure there are no missing critical tasks or dependencies.
            - Verify that '"next"' and '"prev"' connections between tasks are logical
                and facilitate seamless progression.
        - Identify Inefficiencies:
            - Detect and address unnecessary dependencies, bottlenecks, or redundant
                steps that hinder the workflow's efficiency.

    - Allowed Changes:
        - Modify: Clarify and detail the objectives of tasks with insufficient or
            vague directives to ensure they meet the 'final_goal'.
        - Add: Introduce new tasks with clear, detailed descriptions to fill gaps in
            data or structure.
        - Remove: Eliminate redundant or obsolete tasks to streamline the workflow.

    - Maintain Logical Flow:
        - Reorganize task connections ('"next"' and '"prev"') to enhance parallel
            execution and improve overall workflow efficiency.

2. Output Format
    - If No Changes Are Made:
        - Return an empty JSON object to indicate that no modifications were necessary:
            'json{}'.
    - If Changes Are Made:
        - Return a JSON object containing the updated workflow without including the '"
            data"' fields to optimize token usage. This JSON should only include the
            structural changes (task parameters and connections).
```

### D.4 WORKFLOW UPDATE STRATEGIES

We implemented two different workflow update strategies:

- **Update Concurrently**
  In this approach, when a subtask is completed, it immediately triggers the workflow update function, even if other subtasks are still running. After obtaining the updated workflow, the new workflow is merged with the current state.

  - **Trade-off:** This workflow update strategy runs concurrently with task execution, optimizing running time. However, it can result in unnecessary API calls, as some subtasks still in progress may become redundant or misaligned with the updated workflow.

- **Update After Task Completion**
  In this strategy, when a subtask is completed, no new tasks are allocated immediately. Instead, the system waits for all running subtasks to finish before triggering the workflow update. After the update is completed, new subtasks are allocated based on the updated workflow. This approach reduces unnecessary API calls by batching updates.

  - **Trade-off:** This workflow update strategy reduces unnecessary API calls but increases overall running time, as new subtasks are delayed until the workflow update is complete.

In our paper, all the experiments are obtained by using the second strategy to avoid the waste of API usage.

## E   FRAMEWORK OF THE MULTI-AGENT FRAMEWORK

### E.1   OVERVIEW

The multi-agent framework is designed to execute complex tasks by decomposing them into subtasks, which are managed and executed by individual agents. The framework leverages LLM to generate and update workflows dynamically, ensuring robustness, efficiency, and adaptability.

### E.2   KEY COMPONENTS

1. **Agents**
   - **Role Assignment**
     - **Automatic Role Generation**: Roles are automatically generated by LLM during workflow generation and updates.
     - **Flexibility**: By default, roles are not fixed, allowing the system to adapt to the specific requirements of each task.
     - **Role Constraints**: In scenarios with resource constraints, roles can be explicitly defined to limit the number of agents or types of expertise in prompt.
   - **Subtask Assignment**
     - **Matching Expertise**: Subtasks are assigned to agents whose roles best match the task requirements, ensuring tasks are executed by agents with appropriate skills.
     - **One Agent per Subtask**: Only one agent is assigned per subtask to maintain clarity and responsibility.

2. **Workflow Management**
   - **Workflow Generation**
     - **Initial Workflow**: The LLM generates an initial workflow that outlines all subtasks and their dependencies required to achieve the final goal.
     - **Task Dependencies**: Dependencies are defined to ensure logical progression and to facilitate parallel execution where possible.
   - **Workflow Update Mechanisms**
     - Two strategies are employed for updating the workflow:
       (a) **Update Concurrently**
         * **Trigger**: When a subtask is completed, the workflow update function is triggered immediately, even if other subtasks are still running.
         * **Process**: The updated workflow is obtained and merged with the current state.
         * **Trade-off**: Optimizes running time but may result in unnecessary API calls, as some subtasks still in progress might become redundant after the update.
       (b) **Update After Subtask Completion**
         * **Trigger**: No new subtasks are allocated immediately after a subtask is completed. The system waits for all running subtasks to finish before updating.
         * **Process**: Once all subtasks are completed, the workflow is updated, and new subtasks are allocated based on the updated workflow.
         * **Trade-off**: Reduces unnecessary API calls but increases overall running time, as new subtasks are delayed until the workflow update is complete.
         * **Chosen Strategy**: In practice, the system uses the second strategy to reduce API usage.

3. **Dynamic Restructuring**
   - **Mechanism for Dynamic Workflow Restructuring**
     - **Workflow Update Mechanism**: The system includes a robust workflow update mechanism that continuously monitors the execution status of all subtasks. If a subtask fails or is deemed unsolvable, the system triggers an update process.
     - **Re-evaluation of Workflow**: The system systematically reviews the current workflow, taking into account the unsolvable subtask. It assesses the impact of the failed subtask on all subtasks and the overall goal.

- **Adjusting Dependencies**: The workflow is adjusted by removing or modifying the unsolvable subtask and updating dependencies accordingly. This may involve:
  * **Reassigning Subtasks**: Redirecting subtasks to alternative agents or creating new subtasks that can achieve similar outcomes.
  * **Adding New Subtasks**: Introducing new subtasks that offer alternative solutions or pathways to reach the final goal.
  * **Bypassing Unnecessary Steps**: If possible, restructuring the workflow to bypass the unsolvable subtask without compromising the end objectives.

4. **Task Execution**
   - **Parallelism**
     - **Maximizing Parallel Execution**: The workflow is designed to allow subtasks without dependencies to be executed in parallel, optimizing resource utilization and reducing total execution time.
     - **Dependency Management**: Dependencies are minimized where possible to enhance parallelism.
   - **Dependency Minimization**
     - **Dependency Metric**: The system analyzes the standard deviation of the degree distribution in the task graph to identify and minimize critical dependency points.
     - **Reducing Bottlenecks**: By minimizing unnecessary dependencies, the system reduces potential bottlenecks and enhances robustness.

### E.3 WORKFLOW EXECUTION PROCESS

1. **Initial Workflow Generation**
   - The LLM generates a workflow based on the final goal, decomposing it into subtasks with defined dependencies.

2. **Agent Role Assignment**
   - Agents are assigned roles automatically by the LLM.
   - Subtasks are assigned to agents based on role matching.

3. **Subtask Execution**
   - Agents execute their assigned subtasks.
   - Subtasks are executed in parallel where dependencies allow.

4. **Monitoring and Updates**
   - The system monitors subtask completion statuses.
   - Depending on the update strategy, the workflow is updated either concurrently or after all current subtasks are completed.

5. **Dynamic Restructuring**
   - **Detection**: If a subtask is determined to be insufficient or unsolvable for achieving the requirement, the system detects this during execution.
   - **Re-evaluation of Workflow**: The system reviews the current workflow, assessing the impact of the failed subtask on all subtasks and the overall goal.
   - **Workflow Adjustment**: The LLM restructures the workflow dynamically to adjust other subtasks or redefine dependencies.
   - **Continuity**: This ensures that progress toward the final goal continues without significant delays.

6. **Completion**
   - The process continues until all subtasks are completed and the final goal is achieved.

## F LIMITATION AND FUTURE WORK

Although we have generated multiple candidate workflows and selected the one with the highest modularity, it is still not the most efficient. With sufficient computing and data resources, a model trained

specifically for workflow management could significantly enhance the framework's performance. For instance, the LLM could be designed to maximize a reward function centered on key performance indicators such as task completion speed, resource utilization, and minimizing disruptions in the workflow. Such training could lead to the development of more effective workflows. The workflow updater requires global information to function effectively, which can become problematic as the context length increases. This limitation could be addressed by employing a rig or a hierarchical approach to more precisely identify errors or areas lacking efficiency, thereby facilitating more targeted updates and improvements within the workflow.

## G    PILOT STUDY ON AUTO-VALIDATION

In complex workflows involving numerous subtasks, accurately evaluating the correctness of each task becomes challenging, especially when relying solely on a central workflow updater due to extensive context. Recognizing this challenge, we introduced an automated validation mechanism akin to unit testing to enhance task-level correctness and system efficiency.

Specifically, we integrated an auto-validation loop into the Flow system to systematically verify each agent's output immediately after task completion. For computational subtasks involving Python scripts, this validation mechanism generates unit test cases to programmatically verify output correctness. In contrast, for textual or descriptive outputs, a validator LLM agent is employed to assess whether the outputs adequately fulfill the subtask specifications. This validator generates targeted feedback highlighting concerns.

If an output is deemed unsatisfactory, a re-execution agent is invoked, which utilizes the original subtask specification and feedback from the validator to generate a refined solution. Additionally, a history module stores all previous execution attempts alongside their corresponding feedback, guiding subsequent re-execution agents to avoid repeating past errors. Empirically, this process forms a re-execution cycle that systematically improves individual subtask outcomes. To prevent infinite cycles of re-execution, a validation threshold is used, analogous to early stopping methods in optimization tasks. Subtasks are marked as completed upon successful validation, whereas repeated failures beyond a predefined threshold result in the subtask being marked as failed and handled by the workflow updater.

We evaluated this enhanced validation approach in a pilot study involving game development tasks, LaTeX Beamer generation tasks, and a website design task. Given the same backbone model, enabling the validation feature demonstrated significant improvements in individual subtask and task output quality (see Figures 11 and 12). For example, a runnable game combining Tetris and Bejeweled could be generated with auto-validation but failed without it using the backbone model o3-mini-high.

For future work, it would be valuable to perform a rigorous theoretical analysis to better understand the underlying mechanisms contributing to these performance gains. Such analysis could further inform the optimization of validation strategies and workflow design in complex multi-agent workflows.

## H    PROOF OF THEOREM 1

*Proof.* We will compare the expected number of successfully completed subtasks in both workflows.

**Definitions**:

- Let $P_A(v)$ and $P_B(v)$ denote the probability that subtasks $v$ is successfully completed in Workflow A and Workflow B, respectively.

- For each subtasks $v$, let $D_A(v)$ and $D_B(v)$ be the sets of immediate predecessors of $v$ in Workflow A and Workflow B, respectively.

**Success Probability of a subtasks**: In Workflow A, the success probability of subtasks $v$ is given by:

$$P_A(v) = (1 - p_f) \times \prod_{i \in D_A(v)} P_A(i). \tag{1}$$

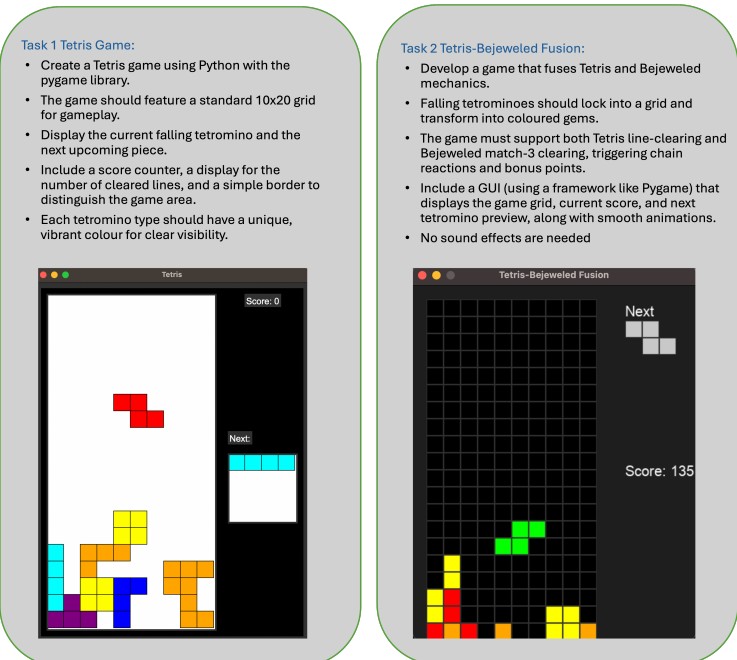

Figure 11: By incorporating auto-validation, Flow is able to automatically generate more sophisticated programs without errors.

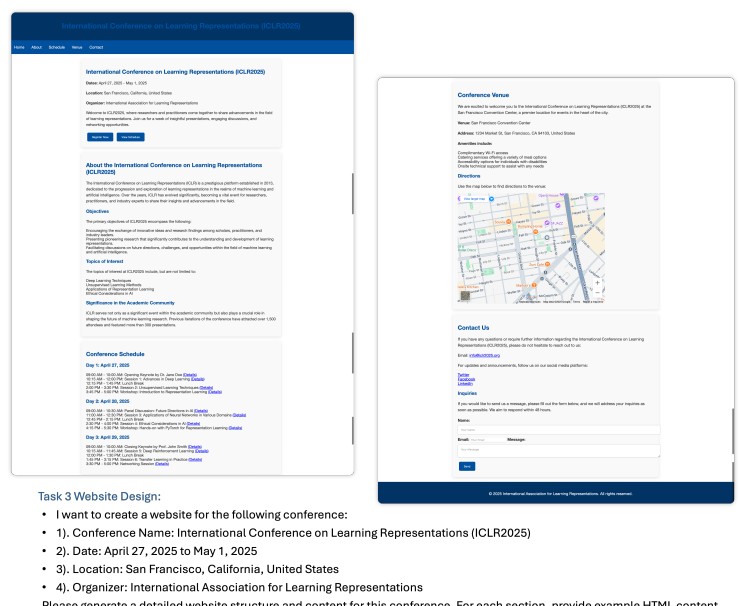

Figure 12: By incorporating auto-validation, Flow can automatically generate more accurate and well-designed websites from the same prompt.

Similarly, in Workflow B:

$$P_B(v) = (1 - p_f) \times \prod_{i \in D_B(v)} P_B(i). \tag{2}$$

**Base Case**: Since the subtasks $v$ with no dependencies (i.e., $D_A(v) = D_B(v) = \emptyset$) have the same success probability in both workflows:

$$P_A(v) = P_B(v) = 1 - p_f.$$

**Inductive Step**: We proceed by induction on the subtasks' dependency levels.

**Comparison for Subtasks $v^*$**: Subtasks $v^*$ has an additional dependency $d$ in Workflow B. Therefore:

$$D_B(v^*) = D_A(v^*) \cup \{d\}.$$

Using equations equation 1 and equation 2, we have:

$$P_A(v^*) = (1 - p_f) \times \prod_{i \in D_A(v^*)} P_A(i),$$

$$P_B(v^*) = (1 - p_f) \times \prod_{i \in D_B(v^*)} P_B(i) = (1 - p_f) \times P_B(d) \times \prod_{i \in D_A(v^*)} P_B(i).$$

Since $D_A(v^*) = D_B(v^*) \setminus \{d\}$, and $P_A(i) = P_B(i)$ for all $i \neq v^*$ (because their dependencies are the same), it follows that:

$$P_B(v^*) = P_A(v^*) \times P_B(d).$$

Because $0 < P_B(d) = P_A(d) < 1$ (since $p_f > 0$), we have:

$$P_B(v^*) = P_A(v^*) \times P_A(d) < P_A(v^*).$$

**Success Probabilities for Other Subtasks**: For all subtasks $v \neq v^*$, $D_A(v) = D_B(v)$, so:

$$P_A(v) = P_B(v).$$

**Expected Number of Successfully Completed Subtasks**: The expected number of successfully completed subtasks in each workflow is:

$$E[S_A] = \sum_{v \in \mathcal{T}} P_A(v),$$

$$E[S_B] = \sum_{v \in \mathcal{T}} P_B(v).$$

Substituting the above findings:

$$
\begin{aligned}
E[S_B] &= \sum_{v \neq v^*} P_B(v) + P_B(v^*) \\
&= \sum_{v \neq v^*} P_A(v) + P_B(v^*) \\
&= \left( \sum_{v \in \mathcal{T}} P_A(v) - P_A(v^*) \right) + P_B(v^*) \\
&= E[S_A] - (P_A(v^*) - P_B(v^*)).
\end{aligned}
$$

Since $P_B(v^*) < P_A(v^*)$, the difference $\Delta P = P_A(v^*) - P_B(v^*) > 0$. Thus,

$$E[S_B] = E[S_A] - \Delta P < E[S_A].$$

Therefore, the expected number of successfully completed subtasks in Workflow A is strictly greater than in Workflow B:

$$E[S_A] > E[S_B].$$

$\square$

