# OpenReview forum: "Flow: Modularized Agentic Workflow Automation"
_ICLR.cc/2025/Conference — ICLR 2025 Poster_

### Official Review · Reviewer_eHFH · 2024-10-25

**Soundness:** 3
**Presentation:** 3
**Contribution:** 3
**Rating:** 8
**Confidence:** 3

**Summary:**

This paper focuses on enhancing large language model (LLM)-based multi-agent systems by improving dynamic workflow management and modular task execution. The proposed system refines workflows using an Activity-on-Vertex (AOV) graph structure, enabling flexible and dynamic task allocation adjustments. The system encourages modularity, allowing for concurrent task execution and minimizing inter-task dependencies. This results in greater robustness and efficiency, especially in complex, evolving task scenarios. Experimental results across tasks like game development, LaTeX slide creation, and website building show that the system can outperform several existing approaches (AutoGen, CAMEL, MetaGPT), with better success rates and human satisfaction scores.

**Strengths:**

This paper introduces dynamic workflow updating and modularization, which appear to be novel.

This paper clearly outlines the methodological innovations, with well-structured sections on each component, from workflow initialization to modular design.

This paper has clear contribution to the design of multi-agent collaboration frameworks, particularly for complex adaptive tasks. By improving modularity and enabling efficient concurrent sub-task execution, the proposed framework has potential applications across fields requiring adaptable, scalable automated task execution.

**Weaknesses:**

While this paper presents some interesting research ideas and promising experiment results, I have several concerns as detailed below.

1. The general idea of supporting dynamic workflow management has been studied previously, in particular in the context of single-agent and multi-agent planning systems. It might be novel to expand this idea in LLM-based multi-agent systems, however the corresponding technical novelty may need to be clearly highlighted and better justified.

2. It seems that a key assumption of this paper is that a highly modular workflow structure can reliably cope with task failures. While this is intuitively meaningful, what happens if we find that a module is unsolvable? In which case, we may need to adjust other modules and potentially the entire workflow structure. Consequently, most part of the workflow execution may be interrupted. Hence, it is essential to theoretically analyze this assumption and understand the specific conditions for it to be valid for practical applications.

3. If building a modular workflow is motivated by the goal of improving robustness of workflow execution, why didn't authors directly analyze the expected amount of changes required by any workflow due to unexpected/random task failures? This expected quantity can be used to determine which workflow should be adopted, in addition to the parallelism metric or the dependency metric. Meanwhile, past studies in the distributed computing community may have developed some existing metrics to measure parallelism and the level of dependency. It is important to clarify why the authors chose to use their own metrics and highlight the corresponding technical novelty.

4. The discussion of how subtasks are assigned to different agents with varied roles is quite brief. The lack of sufficient technical details makes it hard for me to understand this aspect of the system design and its importance to the overall effectiveness of the system. Similarly, while discussing the regeneration of the workflow graph (on page 6), shouldn't this process consider the execution status of the existing/current workflow (e.g., the completion status of different modules) to avoid unnecessary resource wastage? This aspect was not clearly explained in the paper.

5. I don't fully agree with the authors' statement that non-coding tasks may introduce bias on page 7. Besides coding related tasks, many other tasks such as logic reasoning and mathematics analysis tasks may also be important for us to understand the effectiveness and broad usefulness of the newly developed multi-agent system. Meanwhile, given how errors were simulated in Section 5, I was wondering how likely that any task may fail in practice. If tasks seldom fail, is it necessary to find a modular workflow in the first place?

6. Additionally, in terms of the evaluation criteria, I am not sure whether using success rate and human rating is sufficient and fair for comparing different approaches. For each benchmark problem, it is not clear whether the authors followed the common criteria to measure success rates. Meanwhile, for human rating, it largely depends on human participants involved in this research and may not objectively or accurately reveal the varied usefulness of all competing approaches.

**Questions:**

The paper assumes that a highly modular workflow structure can cope with task failures, but what happens if a module turns out to be unsolvable? Does the system have a mechanism to adjust other modules or restructure the entire workflow dynamically? Could you provide more theoretical analysis on the conditions under which modular workflows remain effective in practical applications?

Given the goal of improving workflow robustness, why did you not directly analyze the expected amount of changes required when tasks fail? How could this expected quantity complement the metrics of parallelism and dependency?

Could you elaborate on how subtasks are assigned to different agents with varied roles? How does the assignment strategy impact the system’s overall effectiveness? Also, when regenerating the workflow graph, does the system consider the execution status of current tasks to avoid resource wastage?

Why do you believe non-coding tasks might introduce bias in evaluation? Would it be possible to include additional tasks such as logic reasoning or mathematical analysis to demonstrate the system’s broader applicability?

Could you clarify how human rating was conducted and address concerns that it may not fully reflect the varied usefulness of all competing approaches?

**Details Of Ethics Concerns:**

I do not have ethics concerns.

---

> ### Author Response · Authors · 2024-11-24
> **Response from Authors**
>
> **Q1. The general idea of supporting dynamic workflow management has been studied previously, in particular in the context of single-agent and multi-agent planning systems. It might be novel to expand this idea in LLM-based multi-agent systems.**
>
>
> A1. Conventional work in single-agent and multi-agent planning systems often focuses on training machine learning-based agents using rewards obtained from well-defined environments. In contrast, our LLM-based multi-agent system does not rely on such training processes. Instead, it focuses on effectively leveraging the reasoning capabilities of LLMs to achieve different goals. This shift introduces a unique set of issues that were not typically encountered in earlier approaches.
>
> For example: 1). how to design a unified multi-agent system that enables the generation and updating of workflows for diverse tasks;  2). How to reduce and handle errors during workflow generation and updates, such as output format inconsistencies or careless mistakes that large models might produce during workflow refinement; 3). how to minimize response time and reduce the number of tokens required to achieve a given task.
>
> Addressing these issues is central to our work and should be important for advancing the effectiveness of LLM-based multi-agent systems in dynamic, real-world environments.
>
>
> ---
>
>
> **Q2. What happens if a module turns out to be unsolvable? Does the system have a mechanism to adjust other modules or restructure the entire workflow dynamically?**
>
> A2. Thank you for your insightful question. When a module turns out to be unsolvable, our system is designed to dynamically adjust and restructure any part of the workflow to accommodate this change. This dynamic restructuring is one of the major contributions of our method, and to the best of our knowledge, no existing work offers this capability.
>
> **Mechanism for Dynamic Workflow Restructuring:**
> - **Workflow Update Mechanism:** Our method includes a robust workflow update mechanism that continuously monitors the execution status of all subtasks. If a subtask fails or is deemed unsolvable, the system triggers an update process.
> - **Re-evaluation of Workflow:** The system systematically reviews the current workflow, taking into account the unsolvable module. It assesses the impact of the failed module on all subtasks and the overall goal.
> - **Adjusting Dependencies:** The workflow is adjusted by removing or modifying the unsolvable module and updating dependencies accordingly. This may involve:
> - **Reassigning Tasks:** Modify the subtask to a new subtask that achieves similar results and re-execute the task.
> - **Adding New Modules:** Introducing new modules that offer alternative solutions or pathways to reach the final goal.
> - **Bypassing Unnecessary Steps:** If possible, restructuring the workflow to bypass the unsolvable module without compromising the end objectives.
>
>
> We have provided the full prompt for updating in Appendix D.3.
>
>
>
> ---
>
> **Q3.  Given the goal of improving workflow robustness, why did you not directly analyze the expected amount of changes required when tasks fail? How could this expected quantity complement the metrics of parallelism and dependency?**
>
>
> A3. Thanks for the great point. The goal of improving workflow robustness is not just to minimize the expected amount of changes required when tasks fail but also to reduce task dependencies. Lower dependencies make it easier for LLMs to understand and update the workflow efficiently. For example, even if two workflows have the same expected amount of changes, we would prioritize the one with lower dependencies to enhance interpretability and maintainability.
> Additionally, calculating the expected change count requires estimating the failure probabilities for each task and the impact of each task's failure on the workflow. Accurately determining these probabilities can be challenging, especially in dynamic or unpredictable environments.  However, if reliable estimations were possible, ECC could indeed serve as a valuable complementary metric.

---

> ### Author Response · Authors · 2024-11-24
> **Response from Authors**
>
> **Q4. Meanwhile, past studies in the distributed computing community may have developed some existing metrics to measure parallelism and the level of dependency. It is important to clarify why the authors chose to use their own metrics and highlight the corresponding technical novelty.**
>
> A4. Thank you for the great advice to further help highlight the technical novelty.  We have added the below discussion in Appendix C.
> While there are established metrics in the distributed computing community for assessing parallelism and dependency, we found that they are not directly applicable to our specific context:
>
> 1. **Parallelism Metrics:**
> - **Speedup $S = \frac{T_1}{T_p}$**: This metric measures the ratio of execution time on a single processor $T_1$ to that on multiple processors $T_p$. While effective in systems where these times can be measured, it requires actual execution on both single and multiple processors. In our case, such execution times are not readily obtainable because our focus is on task-solving workflows rather than on processing workloads that can be easily benchmarked in this way.
> - **Amdahl's Law $S(p) = \frac{1}{f_s + \frac{1 - f_s}{p}}$** and **Gustafson's Law $S(p) = p - f_s \cdot (p - 1)$**: Both laws require knowledge of $f_s$, the proportion of the task that is inherently serial, and $p$, the number of processors. Our task graphs have complex dependency structures where tasks cannot be neatly categorized as strictly "serial" or "parallel." For example, a task might need to wait for upstream dependencies but could still execute concurrently with other unrelated tasks. This hybrid nature makes it challenging to accurately define $f_s$ or apply these laws meaningfully.
>
> 2. **Dependency Metrics:**
> - **Cyclomatic Complexity $CC = E - N + p$**: Cyclomatic Complexity measures the number of linearly independent paths through a program, providing an overall complexity measure. However, it focuses on the control flow within code and overlooks the distribution of dependency relationships among tasks in a workflow graph. It does not capture the "dependency dispersion", which are crucial for understanding the impact of dependencies on workflow robustness and the ease with which Large Language Models (LLMs) can comprehend and update the workflow.
>
>
> Given these limitations, we use two simple metrics in our LLM-based multi-agent workflows: 1). **Parallelism Metric**: This metric does not rely on execution time measurements or require assumptions about tasks being strictly serial or parallel. It directly reflects the workflow's potential for concurrent task execution, making it more applicable to our scenario. 2). **Dependency Metric**: We focus on the "dependency dispersion" by analyzing the standard deviation of the degree distribution in the task graph. This metric provides an intuitive reflection of critical dependency points within the workflow. By highlighting how dependencies are distributed among tasks, it helps us understand and mitigate potential bottlenecks, enhancing both robustness and the LLMs' ability to process workflow updates efficiently.
>
> ---
>
> **Q5.  Could you provide more theoretical analysis on the conditions under which modular workflows remain effective in practical applications?**
>
> A5. The results can be derived when comparing two workflows with identical task failure rates:
>
> - **Workflow A (Low Complexity and High Modularity):**
> - Tasks are designed with a few dependencies.
> - The workflow has a simple topology where tasks can be executed in parallel.
> - Each task failure affects only that specific task and does not propagate extensively.
>
> - **Workflow B (High Complexity and Low Modularity):**
> - Tasks have numerous dependencies, creating a complex network.
> - The workflow has a deep topology with many sequential tasks.
> - A single task failure can impact multiple downstream tasks due to high interconnectivity.
>
> **Given that both workflows have the same failure rate for each individual task**, their effectiveness differs significantly due to their structure:
> - **Impact on Effectiveness:**
> - **Workflow A** is more effective on average because its low complexity and high modularity ensure that failures are isolated.
> - **Workflow B** is less effective on average because its high complexity increases the risk of cascading failures. A single task failure can require extensive adjustments or even halt the entire workflow.
> Thank you for raising this excellent point. We believe that deriving theoretical results to enhance the effectiveness of multi-agent systems under different assumptions is a highly meaningful endeavor. We plan to further explore this direction in our future work.
>
>
> ---

---

> > ### Author Response · Authors · 2024-11-24
> > **Response from Authors**
> >
> > **Q7. While discussing the regeneration of the workflow graph (on page 6), shouldn't this process consider the execution status of the existing/current workflow (e.g., the completion status of different modules) to avoid unnecessary resource wastage?**
> >
> > A7. Thanks for the insightful question. Yes, we have avoid the unnecessary resource wastage. Following your constructive comments, we have added the detail of workflow update strategies in Appendix D.4:
> > We implemented two different workflow update strategies:
> > 1. **Update Concurrently:**
> > - In this approach, when a task is completed, it immediately triggers the workflow update function, even if other tasks are still running.
> > - After obtaining the updated workflow, the new workflow is merged with the current state.
> > - **Trade-off:** This workflow update strategy runs concurrently with task execution, optimizing running time. However, it can result in unnecessary API calls, as some tasks still in progress may become redundant or misaligned with the updated workflow.
> > 2. **Update After Task Completion:**
> > - In this strategy, when a task is completed, no new tasks are allocated immediately. Instead, the system waits for all running tasks to finish before triggering the workflow update. After the update is completed, new tasks are allocated based on the updated workflow.
> > - This approach reduces unnecessary API calls by batching updates.
> > - **Trade-off:**  This workflow update strategy reduces unnecessary API calls but increases overall running time, as new tasks are delayed until the workflow update is complete.
> > In our paper, all the experiments are obtained by using the second strategy to avoid the waste of API usage.
> >
> > ---
> >
> >
> > **Q8. I don't fully agree with the authors' statement that non-coding tasks may introduce bias on page.  Besides coding related tasks, many other tasks such as logic reasoning and mathematics analysis tasks may also be important for us to understand the effectiveness and broad usefulness of the newly developed multi-agent system.**
> >
> >
> > A8. Thank you for your valuable comment. We fully agree with your observation, and we acknowledge that this was a writing issue in the paper. What we meant to convey is that, in our setup, all outputs generated by different multi-agent frameworks are represented as code rather than natural language. To avoid confusion, we have removed the original sentence from the paper.
> >
> > Additionally, we have incorporated the commonly used GSM8K benchmark for mathematical reasoning using GPT-4o-mini. CoT refers to the 'chain of thought' approach (NeurIPS'22), while Direct indicates reasoning without employing the chain of thought method. The results are presented below:
> >
> > | **Dataset** | **Direct** | **CoT** | **Ours** |
> > |-------------|------------|---------|----------|
> > | GSM8K       | 92.7       | 92.4    | 93.0     |
> >
> >
> > It is important to note, however, that the mathematical questions in this dataset are relatively simple. As described in the GSM8K dataset documentation, "These problems take between 2 and 8 steps to solve, and solutions primarily involve performing a sequence of elementary calculations using basic arithmetic operations (+, −, ×, ÷). A bright middle school student should be able to solve every problem." Given this level of complexity, we believe that using multiple agents for this benchmark does not provide a significant advantage. This is why we initially refrained from using such commonly available benchmarks in our evaluation. This limitation of the benchmark was also noted by Reviewer eHFH.
> >
> > ---

---

> > > ### Author Response · Authors · 2024-11-24
> > > **Response from Authors**
> > >
> > > **Q9. Meanwhile, given how errors were simulated in Section 5, I was wondering how likely that any task may fail in practice. If tasks seldom fail, is it necessary to find a modular workflow in the first place?**
> > >
> > >
> > > A9. Based on your constructive comments, we have reported the updates in the following tables for all tasks using GPT-4o-mini. In this experiment, each task was run five times on different models, and the average of the results was calculated as the final outcome. Specifically, we evaluate the workflows using three metrics:
> > >
> > > - #Initial Tasks: The number of tasks identified within the workflow after selecting the optimal workflow but before execution begins.
> > >
> > > - #Changed Tasks: The number of tasks in the initial workflow that was updated after completing the workflow execution.
> > >
> > > - Changed Ratio: Calculated as the average number of changed tasks divided by the average number of initial tasks, providing an intuitive measure of the proportion of tasks updated during execution.
> > >
> > > We ran each task five times and reported the average values in the table below.
> > >
> > >
> > > | **LLM-agent** | **Task**            | **#Initial Task (avg.)** | **#Changed Tasks (avg.)** | **Changed Ratio (avg.)** |
> > > |---------------|---------------------|--------------------------|---------------------------|--------------------------|
> > > | GPT-4o-mini   | Gobang Game         | 8                        | 2.8                       | 35%                      |
> > > |               | Website Design      | 7.2                      | 3.4                       | 47%                      |
> > > |               | LaTeX Beamer Writing | 9.2                      | 4.8                       | 53%                      |
> > >
> > >
> > > Additionally, we have included a lower-performance model, GPT-3.5-turbo, in our evaluation. As expected, GPT-3.5-turbo required more updates during runtime as expected. This is because GPT-3.5-turbo has comparatively weaker task execution capabilities, resulting in more frequent workflow adjustments due to insufficiently generated data. Notably, our method outperformed the best baseline on GPT-3.5-turbo by **29.9\%**, **26.4\%**, and **19.2\%** on quantitative measures across the three tasks,  respectively (Appendix B.5). This further highlights the effectiveness of our workflow generation and updating framework.
> > >
> > >
> > > ---
> > >
> > > **Q10. I am not sure whether using success rate and human rating is sufficient and fair for comparing different approaches. Meanwhile, for human rating, it largely depends on human participants involved in this research and may not objectively or accurately reveal the varied usefulness of all competing approaches.**
> > >
> > >
> > > A10. Thank you for your valuable question,  this should not be a cocern.
> > >
> > > The success rate is evaluated as a binary metric, where each task is classified as either a "success" (score of 1) or a "failure" (score of 0). For instance, in the coding task benchmark, if the generated code executes successfully and produces the correct output, it is considered a success (score 1). Otherwise, it is marked as a failure (score 0). This approach provides an objective and consistent measure of task completion. We strictly adhered to established criteria for each benchmark problem to ensure fair comparison across different approaches.
> > >
> > > For human evaluation, we randomly selected 50 participants who have a background in reinforcement learning. This ensures that the evaluation is statistically meaningful and that participants possess the expertise needed to provide informed judgments. Additionally, to minimize bias and ensure objectivity, these participants were not involved in this research.

---

> ### Comment · Reviewer_eHFH · 2024-11-25
> **Thank you for the rebuttals**
>
> Thank the authors for providing rebuttals.
>
> For the response to Q2 and Q3, can the authors please provide additional experiment results to support their arguments? In particular, I would hope to see detailed results to show the effectiveness of each step of the dynamic workflow restructuring mechanism. It would also be great to see how the interpretability and maintainability could be enhanced for a workflow with the same expected amount of changes, given the metrics of parallelism and dependency.
>
> For the response to Q5, can the analysis be conducted mathematically rather than intuitively?
>
> For the two strategies mentioned in the response to Q7, do you have more experiment results regarding the extra cost in terms of API calls, especially for the first strategy? Does the strategy help improve performance at the expense of more API calls, why or why not?
>
> For the response to Q10, it would be important to know how the participants were recruited for the empirical study (e.g., How to determine whether a participant has sufficient/suitable background in reinforcement learning? Did you consider other background for each participant? Did any participants know either directly or indirectly any researchers involved in this work? Were any participants given an introduction to the reported study before or after experiments?). How were the experiments conducted to minimize bias and ensure objectivity (e.g., the detailed guidance provided to the participants during the experiments).

---

> > ### Author Response · Authors · 2024-11-25
> >
> > Dear Reviewer eHFH,
> >
> > We are following your comments and preparing the additional information. The mathematical analysis is also feasible to provide, sincerely thank you for helping strengthen our technical contribution. We will update our response shortly.
> >
> > Best,
> >
> > The Authors

---

> ### Author Response · Authors · 2024-11-30
> **Followup Response from Authors**
>
> **Dear Reviewer eHFH,**
>
> ### Thank you so much for all the constructive comments. We apologize for sending our response during Thanksgiving and the weekend. We have provided followup response as follows.
>
> ---
>
> **Followup Q1. I would hope to see detailed results to show the effectiveness of each step in Q2 of the dynamic workflow restructuring mechanism.**
>
> Thank for your question. We provided to examples in the following.
>
>
>  ###  Example 1: Website
>
> > **Website Task Requirement**:
> I am designing a website for the International Conference on Learning Representations (ICLR2025), which will take place from April 27, 2025, to May 1, 2025, in San Francisco, California, United States. The conference is organized by the International Association for Learning Representations.
>
> > Note that:
> 1). For each section, I would like to see example HTML content. Additionally, a sample
>       CSS stylesheet should be provided to style the website. The content must be
>      professional, clear, and appropriate for an international academic conference.
> 2). The website should include all the provided details, including a comprehensive
>      conference schedule and a section dedicated to the conference venue, featuring a
>      map.
>
>  ### Before Updating:
> 0.  "Collect and organize all relevant conference details, including dates, location, and organizing body information in HTML."
>    *Child nodes:* subtask 1, subtask 3, subtask 4
> 1.  "Draft example HTML content for the website's homepage, including a brief introduction to the conference and key highlights."
>    *Child node:* subtask 5
> 2.  "Create a sample CSS stylesheet to ensure the website has a professional and cohesive design."
>    *Child node:* subtask 5
> 3.  "Develop the conference schedule section in HTML, detailing all sessions, speakers, and timings."
>    *Child node:* subtask 5
> 4.  "Design the conference venue section in HTML, including details of the venue and an embedded map of San Francisco."
>    *Child node:* subtask 5
> 5.  "Integrate HTML content and CSS stylesheet, ensuring all sections are styled and functional."
>
> ### After Updating:
> 0.  "Collect and organize all relevant conference details, including dates, location, and organizing body information in HTML."
>    *Child nodes:* subtask 1, `subtask 2`, subtask 3, subtask 4
> 1.  "Draft example HTML content for the website's homepage, including a brief introduction to the conference, key highlights, registration information, and key dates."
>    *Child node:* subtask 5
> 2.  `"Create a sample CSS stylesheet to ensure the website has a professional and cohesive design, incorporating specific design elements and color schemes."`
>    *Child node:* subtask 5
> 3. `"Develop the conference schedule section in HTML, detailing all sessions, speakers, timings, and session descriptions for clarity and completeness."`
>    *Child node:* subtask 5
> 4.  "Design the conference venue section in HTML, including details of the venue, an embedded map of San Francisco, and additional venue information."
>    *Child node:* subtask 5
> 5.  "Integrate HTML content and CSS stylesheet, ensuring all sections are styled and functional."
>
>
> Adjusting Dependencies: Originally, the `subtask 2` "Create a sample CSS stylesheet to ensure the website has a professional and cohesive design" was `executed at the same time` as `subtask 0` "Collect and organize all relevant conference details, including dates, location, and organizing body information." However, this was not a logical sequence, as we did not have the necessary HTML template information to create the corresponding CSS. During excutation, our systems adjust the subtasks Dependencies, moving the `subtask 2` to act as a `child node` of `subtask 0`. These adjustments allowed an more accurate responds generated
>
> Reassigning Tasks: we can see in the workflow, the task description is changed to  more precise and accurate, direct answering the requirement. for example  `subtask 3` `"Develop the conference schedule section in HTML, detailing all sessions, speakers, and timings."`  was changed to `"Develop the conference schedule section in HTML, detailing all sessions, speakers, timings, and session descriptions for clarity and completeness."`

---

> ### Author Response · Authors · 2024-11-30
> **Followup Response from Authors**
>
> ### Example 2: LaTex Beamer
> > **LaTex Beamer Task  Requirement**: I am a lecturer teaching a machine learning course to research students, I am preparing lecture slides on various reinforcement learning algorithms.
>
> > Note that: 1). Given that the lecture duration is 2 hours, the slides should span approximately
>      30 pages.
> 2). For each reinforcement learning algorithm covered, the slides will include the
>      following key components: the motivation behind the algorithm, the problem it aims
>       to solve, an intuitive solution, and the detailed mathematical equations that
>      underpin the method.
> 3).  It is essential that the lecture is comprehensive and self-contained, providing
>      students with a clear understanding of each algorithm from both a conceptual and
>      technical perspective.
>
>
> ###  Before Updating:
> 1.  "Research and select the key reinforcement learning algorithms to be covered in the lecture, ensuring a balanced representation of popular methods."
>    *Child nodes:* subtask 2
> 2.  "Create the LaTeX structure for the lecture slides, including title slide, agenda, and formatting for 30 pages."
>    *Child nodes:* subtask 3
> 3.  "For each selected reinforcement learning algorithm, generate content for the slides that includes motivation, problem statement, intuitive solution, and detailed mathematical equations."
>
>
> ### After Updating:
> 1.  "Research and select the key reinforcement learning algorithms to be covered in the lecture, ensuring a balanced representation of popular methods."
>    *Child nodes:* subtask 2
> 2.  "Create the LaTeX structure for the lecture slides, including title slide, agenda, and formatting for 30 pages. `Ensure placeholders for each algorithm's motivation, problem, solution, and equations are included.`"
> *Child nodes:* subtask 3
> 3. "For each selected reinforcement learning algorithm, generate detailed slide content that includes motivation, problem statement, intuitive solution, and detailed mathematical equations. `Ensure the lecture is self-contained and content is structured to fill 30 slides.`"
>    *Child nodes:* subtask 4
>
> 4.  `"Compile all generated content into the LaTeX document, ensuring proper referencing and self-contained explanations. Verify the document's completeness, coherence, and balanced distribution across 30 slides. Ensure each algorithm is covered in detail with motivation, problem, solution, and equations."`
>
> Adding New Modules: In the updated workflow, a new subtask (subtask 4) was added: `"Compile all generated content into the LaTeX document, ensuring proper referencing and self-contained explanations. Verify the document's completeness, coherence, and balanced distribution across 30 slides. Ensure each algorithm is covered in detail with motivation, problem, intuitive solution, and equations."` This module was introduced to ensure a final quality check and integration of the generated content into the lecture slides.
>
> Reassigning Tasks: `subtask 3` was reassigned with a refined description to ensure the output aligns with more detailed and structured requirements.  Originally, it only focused on generating slide content for selected algorithms, but in the updated workflow, it further emphasize `Ensure the lecture is self-contained and content is structured to fill 30 slides.` for achieve the `30 slides task requirement`.
>
>
> ### Example 3: error handling
>
> Additionally, for the Re-evaluation of Workflow and Bypassing Unnecessary Steps in cases of unsolvable subtasks, this capability is demonstrated in the error handling mechanism described in Section 5. We simulate scenarios such as unstable internet connections or internal API errors (e.g., ChatGPT occasionally returning "there is an error, failed to generate a response"). The results show that when a subtask's status becomes inconsistent with the generated data, our update mechanism effectively adds new, similar tasks while bypassing unnecessary steps associated with failed subtasks.
>
> For example, in the Gobang Game workflow, if subtasks  `Design the Game UI` and `Implement Naive AI` fail, the workflow automatically re-evaluates the process. This leads to the  `UI Module Redesign` and  `the Reimplementation of AI` with updated dependency,  while removing the old, failed subtasks that are no longer necessary.

---

> ### Author Response · Authors · 2024-11-30
> **Followup Response from Authors**
>
> **Followup Q2. It would also be great to see how the interpretability and maintainability could be enhanced for a workflow with the same expected amount of changes, given the metrics of parallelism and dependency.**
>
> Followup A2. We believe there may be a misunderstanding regarding the term interpretability in our response. The interpretability we refer to does not pertain to human interpretability but rather to the simplicity of workflows. Specifically, a JSON-formatted workflow containing numerous dependencies and large amounts of data might be challenging for large models to process and update effectively. Therefore we reduce this complexity and reduce dependencies, making workflows easier for large models to understand the subtasks relations during updates.
>
> For maintainability, we conduct an ablation study.  We measures the average proportion of subtasks indirectly affected (and requiring modification) by changes in a workflow with the **Propagation Rate** .  Workflows with high propagation rates are hard to scale since changes in one part of the system are less likely to disrupt others.
> Specifically, the **Propagation Rate** it is defined as:
> $$
> \text{Propagation Rate} = \frac{1}{\text{Number of Updated Subtasks}} \sum \left( \frac{\text{Number of Indirectly Affected Subtasks}}{\text{Total Number of Subtasks in Workflow (per update)}} \right)
> $$
> 1. **Numerator of the Sum**: $\text{Number of Indirectly Affected Subtasks}$: The count of subtasks influenced as a result of a change in other tasks.
> 2. **Denominator of the Fraction**: $\text{Total Number of Subtasks in Workflow (Per Update)}$: The total number of subtasks in the workflow after the update, which serves as the baseline for normalization.
> 3. **Averaging Over Updated Subtasks**:  This ensures the Propagation Rate reflects the average number of indirectly affected subtasks per updated subtask (same amount of changes).
>
> ### Impact of Dependencies on Workflow
> #### LaTeX Beamer Workflow
> | Graph Type | Propagation Rate (%) |
> |----------------------|-----------------------|
> | With two metrics | 16 |
> | Without two metrics | 42 |
> #### Gobang Game Workflow
> | Graph Type | Propagation Rate (%) |
> |----------------------|-----------------------|
> | With two metrics | 27 |
> | Without two metrics | 40 |
> #### Website Design Workflow
> | Graph Type | Propagation Rate (%) |
> |----------------------|-----------------------|
> | With two metrics | 34 |
> | Without two metrics | 73 |
> ---
>
>
> **Followup Q3. For the response to Q5, can the analysis be conducted mathematically rather than intuitively?**
>
> Followup A3. By following your constructive comments, we show that introducing additional dependencies in a workflow reduces the expected success rate of subtasks.
> #### **Theorem**
> **(Modular Workflow and Expected Success)**
> Consider two topologically sorted workflows $A$ and $B$, each consisting of $N$ subtasks labeled $t_1, t_2, \ldots, t_N$ in execution order. Let the set of immediate predecessors (dependencies) of a subtask $t_i$ in workflows $A$ and $B$ be denoted by $D_A(t_i)$ and $D_B(t_i)$, respectively. Assume:
> 1. The expected subtask failure rate is $p_f$, where $0 < p_f < 1$.
> 2. Workflow $B$ introduces at least one additional dependency compared to Workflow $A$. Specifically, there exists a subtask $t^*$ such that $D_B(t^*) = D_A(t^*) \cup \{d\}$ for some $d \notin D_A(t^*)$. For all other subtasks $t \neq t^*$, $D_A(t) \subseteq D_B(t)$.
>
> The expected number of successfully completed subtasks in Workflow $A$ is strictly greater than in Workflow $B$.
>
> #### **Proof**
> We compare the expected number of successfully completed subtasks in workflows $A$ and $B$.
> 1. **Probability of Success for a Subtask**
> For a subtask $t$, the probability of successful completion depends on the success of its dependencies:
> $$
> P_A(t) = (1 - p_f) \times \prod_{i \in D_A(t)} P_A(i),
> $$
> $$
> P_B(t) = (1 - p_f) \times \prod_{i \in D_B(t)} P_B(i).
> $$
> 2. **Base Case:**
> For subtasks with no dependencies ($D_A(t) = D_B(t) = \emptyset$):
> $$
> P_A(t) = P_B(t) = 1 - p_f.
> $$
> 3. **Inductive Step:**
> Consider a subtask $t^*$ with an additional dependency $d$ in Workflow $B$:
> $$
> D_B(t^*) = D_A(t^*) \cup \{d\}.
> $$
> The success probabilities become:
> $$
> P_A(t^*) = (1 - p_f) \times \prod_{i \in D_A(t^*)} P_A(i),
> $$
> $$
> P_B(t^*) = (1 - p_f) \times P_B(d) \times \prod_{i \in D_A(t^*)} P_B(i).
> $$
> Since $P_B(d) < 1$, it follows that:
> $$
> P_B(t^*) < P_A(t^*).
> $$
> 4. **Expected Number of Successfully Completed Subtasks:**
> The expected success for workflows $A$ and $B$ is:
> $$
> E[S_A] = \sum_{t \in \mathcal{T}} P_A(t), \quad E[S_B] = \sum_{t \in \mathcal{T}} P_B(t).
> $$
> Substituting the probabilities:
> $$
> E[S_B] = E[S_A] - (P_A(t^*) - P_B(t^*)) < E[S_A].
> $$
> Thus, Workflow $A$ achieves a strictly higher expected number of successfully completed subtasks than Workflow $B$.
>
> Thank you very much for help improving our paper.

---

> ### Author Response · Authors · 2024-11-30
> **Followup Response from Authors**
>
> **Followup Q4. For the two strategies mentioned in the response to Q7, do you have more experiment results regarding the extra cost in terms of API calls, especially for the first strategy? Does the strategy help improve performance at the expense of more API calls, why or why not?**
>
> Followup A4. Thank you for the insightful comments, we follow your comments and added additional experiments for two settings—Update Concurrently and Lazy update (Update After Task Completion) —using two models: GPT-4o-mini and GPT-3.5-Turbo, for comparison. The experiment will involve the following metrics:
>
> 1. API Calls for Change: The total number of times the OpenAI API was called for additional workflow updates during the experiment.
> 2. Score: Measure the average score of all three experimental settings by including all quantitative metrics.
> 3. Completion Time: Record the time taken for the models to complete the tasks under both settings.
>
> ### Table 5: Number of  API Calls vs. Performance
> |  Strategy                             |  API Calls for Change  |  Overall Score (%)  |  Completion Time (s)      |
> |-----------------------------------------|---------------------------|-------------------|-----------------------------|
> | Update Concurrently (GPT- 4o-mini)           | 0.85                      | 95.57               | 66.75 ± 19.11               |
> | Update Concurrently (GPT-3.5-turbo)         | 0.92                      | 76.67               | 33.82 ± 13.45               |
> | Lazy Update  (GPT-4o-mini)  | 0.65                      | 93.33                | 69.27 ± 14.74           |
> | Lazy Update (GPT-3.5-turbo)| 0.68                      | 78.00          | 30.64 ± 12.16           |
>
> we can see for both model Update After Task Completion resulting in the lower number of api calls per task and relatively the same accuracy rate also the completion time is shorter then Update Concurrently.
>
> The experiments show that  Update Concurrently strategy does not show significant improve performance. We believe the reason is that for both methods, all subtasks will be eventually checked.
>
>
> **Followup Q5.  For the response to Q10, it would be important to know how the participants were recruited for the empirical study.**
>
> Followup A5. Thank you for these important questions! We believe they are crucial and should have been made clear. We will carefully incorporate the details below into a future version.
>
> #### **Participant Recruitment and Selection**
> - Participants were selected based on their expertise in reinforcement learning, specifically senior PhD students with teaching experience in reinforcement learning or faculty members specializing in machine learning.
> - The primary consideration was expertise in reinforcement learning, as it was essential for evaluating the study's outcomes. No other specific backgrounds were included, as they were not relevant to the evaluation criteria.
> #### **Measures to Minimize Bias**
> - While participants may have known the researchers directly or indirectly, several measures were implemented to eliminate bias:
> - The method names were completely masked, and results were labeled anonymously as 1, 2, 3, and 4.
> - Participants were blinded to which method corresponded to the researchers’ work.
> - Rankings were based solely on **how well the results aligned with the task requirements**, with no additional information about the study or methods shared before or after the experiments.
> #### **Detailed Guidance for Participants**
> - Participants were provided with clear task requirements and instructions to rank outcomes from high quality to low quality based on their alignment with the requirements.
> - To ensure objectivity, participants evaluated results without knowing their origin, relying strictly on the defined criteria.

---

> ### Author Response · Authors · 2024-11-30
> **Followup Response from Authors**
>
> #### **Quantitative Validation Metrics**
>
> We also employed quantitative metrics to ensure fairness and objectivity.
>
> Below is a concrete example of how these metrics were applied, specifically for the Gobang Game task (as shown in Fig. 1 of our paper).
>
> ---
>
> ### **An Example: Quantitative Metrics for the Gobang Game Task**
> #### **Requirements for the Gobang Game**
> The game includes a naive AI and a user interface. The game should:
> 1. End when either a player wins (five pieces aligned) or the board is filled.
> 2. Clearly indicate whose turn it is.
> 3. Display a message when the game concludes, specifying the winner.
> 4. Allow the user to play as either black or white stones.
> ### **Metrics**
> 1. **Metric 1: Compilable**
> - **Objective:** Verifies whether the game code can compile and run successfully.
> - **Scoring:**
> - If the game compiles successfully → **1 mark**.
> - If the code fails to compile (e.g., termination error) → **0 marks**.
> 2. **Metric 2: Interactive**
> - **Objective:** Ensures that both the human player (black) and the AI (white) can make moves without errors or crashes.
> - **Scoring:**
> - If both players can make moves successfully → **1 mark**.
> - If one or both players cannot make moves → **0 marks**.
> 3. **Metric 3: Game Rule**
> - **Objective:** Assesses whether the game ends correctly when five pieces are aligned (horizontally, vertically, or diagonally).
> - **Scoring:**
> - If the game can ends correctly → **1 mark**.
> - If it does not or wrong in game logic → **0 marks**.
> ---
> ### **Example Evaluation of Methods**
> #### **Autogen:**
> - **Compilable:** The game compiles successfully and runs without errors → **1 mark**.
> - **Interactive:** Both human and AI players can make moves (as shown in Fig. 1 in our paper) → **1 mark**.
> - **Game Rule:** The game does not end correctly when five black pieces are aligned (as shown in Fig. 1 in our paper) → **0 marks**.
> #### **MetaGPT:**
> - **Compilable:** The game compiles successfully → **1 mark**.
> - **Interactive:** Both human and AI players can make moves using "X" or "O" → **1 mark**.
> - **Game Rule:** The game does not follow the rules correctly (it can not check for five aligned pieces and is not a proper Gobang game) → **0 marks**.
> #### **Camel:**
> - **Compilable:** The game compiles successfully → **1 mark**.
> - **Interactive:** There is no AI opponent in the game (only black pieces appear shown in Fig 1 in our paper), so human vs AI play is not possible → **0 marks**.
> - **Game Rule:** The game does not stop correctly  → **0 marks**.
> #### **Flow (ours):**
> - **Compilable:** The game compiles successfully → **1 mark**.
> - **Interactive:** Both human and AI players can make moves → **1 mark**.
> - **Game Rule:** The game ends correctly when five pieces are aligned, and there is a restart button to reset the game → **1 mark**.
> ---
>
> Scores were binary and the evaluation involved no subjective judgment.

---

> > ### Comment · Reviewer_eHFH · 2024-12-01
> > **Thank the authors for the detailed rebuttals**
> >
> > I would like to thank the authors for providing more experiment results and detailed explanation.
> >
> > I will update my score accordingly. Meanwhile, I would like to recommend the authors to incorporate all the extra results and discussions into their revised paper. Thank you!

---

> > > ### Author Response · Authors · 2024-12-01
> > >
> > > Dear Reviewer eHFH,
> > >
> > > Thank you very much for your positive support and for taking the time to help us improve our paper. Your comments are both constructive and detailed, and we sincerely appreciate your responsibility and thoughtfulness. We have carefully organized all the information you provided for inclusion in our paper.
> > >
> > >
> > > All the best wishes,
> > >
> > > Authors

---

### Official Review · Reviewer_uiek · 2024-11-02

**Soundness:** 3
**Presentation:** 2
**Contribution:** 3
**Rating:** 6
**Confidence:** 3

**Summary:**

The paper presents a multi-agent system approach to automating (essentially coding) tasks such as website creation or game creation. Their multi-agent approach does better than other single agent approaches from the literature. A key to tracking the multi-agent system's progress on such complex tasks was utilizing a graphical representation of sub-tasks, task assignments and progress, namely activity-on-vertex directed acyclic graphs to represent the workflow. Results on multiple tasks (while anecdotal) show the proposed approach does better than other approaches in the literature.

**Strengths:**

The paper presents an interesting and intuitive approach to representing workflows and enabling multi-agent systems to execute complex tasks. Results support the validity of the approach. The paper also tackles an important open problem in the literature: automating complex tasks. The authors also perform a good analysis of the experiments, taking the time to specify errors that different agents performed in completing their tasks.

**Weaknesses:**

The authors mostly evaluate their approach anecdotally as opposed to on standard benchmarks or datasets. This is understandable given the lack of complex task datasets in this domain.

The paper contains grammar mistakes and presentation issues that make the paper difficult to read. A few examples: in the abstract, did you mean "define workflows *as* activity on vertex (AOV)"?; line 162 should say "inspired by" as opposed to "inspire by"; Figure 1 should say "the game must end" instead of "must be ended"; Figure 2 should say "initial" as opposed to "intial"; in Tables 1, 2 and 3, instead of saying "ours", say "Flow (ours)" to be consistent with the writeup in the various sections; line 118, "1)" instead of "1)."; line 132, "LLM-based" as opposed to "LLM based".

**Questions:**

NA

---

> ### Author Response · Authors · 2024-11-24
> **Response from Authors**
>
> **Q1. The authors mostly evaluate their approach anecdotally as opposed to on standard benchmarks or datasets. This is understandable given the lack of complex task datasets in this domain.**
>
> A1. Thank you for your insightful comments. We completely agree with your point.  We believe the true potential of a multi-agent system lies in handling complex tasks that require high levels of parallelism to enhance efficiency, as well as tasks that necessitate coordination among agents with expertise in diverse knowledge domains.
> For completeness, we have added two standard benchmarks in our evaluation for completeness. Although the result shows that our system outperforms Chain of Thought reasoning. We believe that the major performance improvement is not due to the use of multi agents but is primarily attributed to the additional checking process embedded in the workflow.
> Additionally, we have conducted additional experiments to evaluate our framework:
> 1. **Using a Lower-Performance Model:**
> - We employed GPT-3.5-turbo for all baselines across three tasks. Our method consistently outperformed the baselines, even when using this lower-performance model.
> 2. **Analyzing Running Time:**
> - Without updating the workflow, our method is much faster than all baselines (which were also tested without workflow updates).
> - When incorporating workflow updates, our method achieved a running time comparable to the best-performing baseline.
>
>
> ---
>
>
> **Q2. The paper contains grammar mistakes and presentation issues that make the paper difficult to read. A few examples: in the abstract, did you mean "define workflows as activity on vertex (AOV)"?; line 162 should say "inspired by" as opposed to "inspire by"; Figure 1 should say "the game must end" instead of "must be ended"; Figure 2 should say "initial" as opposed to "intial"; in Tables 1, 2 and 3, instead of saying "ours", say "Flow (ours)" to be consistent with the writeup in the various sections; line 118, "1)" instead of "1)."; line 132, "LLM-based" as opposed to "LLM based".**
>
>
> A2. Thank you for taking the time and detailed review. We have carefully reviewed and addressed all the points you highlighted:
> - Corrected the abstract to "define workflows as activity on vertex (AOV)."
> - Revised line 162 to say "inspired by" instead of "inspire by."
> - Updated Figure 1 to state "the game must end" instead of "must be ended."
> - Corrected the typo in Figure 2 to "initial" instead of "intial."
> - Updated Tables 1, 2, and 3 to use "Flow (ours)" for consistency with the writeup in the sections.
> - Fixed line 118 to use "1)" instead of "1)."
> - Revised line 132 to "LLM-based" instead of "LLM based."
> We have also proofread the paper to ensure clarity, consistency, and correctness throughout. Thank you again for bringing these issues to our attention.

---

> > ### Comment · Reviewer_uiek · 2024-11-27
> >
> > Thank you for addressing my comments. I do believe it would be useful to quantify the computational time comparisons, although I understand that if you didn't already have those numbers, it would be difficult to include in the response above. I encourage including them in future versions of the paper.

---

> ### Author Response · Authors · 2024-11-30
> **Followup Response From Authors**
>
> Dear Reviewer uiek,
>
>
> Following your advice, we conducted a running time analysis comparing our method (Flow) with other baselines (MetaGPT, CAMEL, and Autogen). The results are provided below and have also been included in the revised version of our paper. This analysis highlights the high efficiency of our workflow implementation, both with and without updates. We believe that our code can serve as a decent code function for further research and development in Multi-agent system.
>
>
> |                 | Task               | Flow (w/o update) | Flow (w/ update) | MetaGPT          | CAMEL           | Autogen         |
> |-----------------|--------------------|-------------------|------------------|------------------|-----------------|-----------------|
> |  GPT-3.5-turbo  |                    |                   |                  |                  |                 |                 |
> |                 | Gobang Game        | 26.12 ± 11.35     | 33.57 ± 12.46    | 34.00 ± 15.12    | 121.52 ± 20.87  | 31.00 ± 14.67   |
> |                 | Conference Website | 23.46 ± 10.84     | 34.23 ± 13.12    | 85.14 ± 18.52    | 41.96 ± 12.89   | 44.00 ± 15.34   |
> |                 | Latex Beamer       | 18.34 ± 9.73      | 24.12 ± 10.89    | 29.92 ± 14.87    | 166.00 ± 22.64  | 31.00 ± 16.78   |
> |  GPT-4o-mini   |                    |                   |                  |                  |                 |                 |
> |                 | Gobang Game        | 60.45 ± 14.78     | 72.34 ± 13.45    | 99.45 ± 16.92    | 110.94 ± 19.67  | 148.72 ± 25.34  |
> |                 | Conference Website | 22.78 ± 12.45     | 52.14 ± 14.89    | 127.49 ± 17.52   | 74.53 ± 18.34   | 86.78 ± 21.23   |
> |                 | Latex Beamer       | 44.21 ± 13.67     | 83.34 ± 15.89    | 66.72 ± 19.45    | 106.34 ± 20.78  | 95.21 ± 22.56   |
>
>
>
> The key reason our system maintains a low time cost is:
> - Unlike existing graph-based systems, we explicitly encourage parallel execution and low dependency by calculating parallelism and dependency metrics for the generated workflows. By selecting multiple candidate graphs based on these quantitative metrics, we directly facilitate the creation of highly parallel and less complex workflows.
> - Although the work could be done in parallel, existing implementations typically do not execute tasks simultaneously but in sequence. Our implementation fully utilizes parallel execution capabilities, significantly reducing the expected execution time. By benefiting directly from parallel processing.
>
>
> Have a great weekend and all the best wishes for a wonderful Thanksgiving!
>
>
> Best,
>
> The Authors

---

### Official Review · Reviewer_RcMF · 2024-11-03

**Soundness:** 2
**Presentation:** 3
**Contribution:** 2
**Rating:** 5
**Confidence:** 4

**Summary:**

This paper introduces a novel approach to enhancing the performance of LLM-based multi-agent systems through the use of activity-on-vertex (AOV) graphs. By defining workflows as AOV graphs, the authors enable dynamic updates during task execution, which facilitates better resource allocation and real-time adjustments. This flexibility is crucial for efficiently managing complex tasks. The paper presents a practical multi-agent framework that incorporates quantitative measures to evaluate workflows, allowing for efficient selection and improved planning capabilities. To empirically demonstrate the framework's effectiveness, the authors conduct experiments across three diverse tasks: game generation, LaTeX slide generation, and website design. They provide detailed evaluations and analyses of the results, showcasing the superior performance of their framework compared to existing open-source solutions.

**Strengths:**

-  The paper effectively points out a significant gap in existing frameworks, specifically regarding the adaptability of workflows in real-time, especially in the face of unforeseen challenges.
-  The paper is well-structured and easy to follow, offering clear definitions and detailed formulas that enhance comprehension.

**Weaknesses:**

-  Limited novelty. While dynamic workflow generation and execution have been extensively discussed in prior studies (e.g., [1][2][3][4]), this paper's approach lacks sufficient novelty. Directed acyclic graphs (DAGs) and graph-based frameworks have already been established as effective structures in LLM-based agent frameworks. To strengthen its contribution, the paper should include comparative analysis with these existing approaches, highlighting specific advantages and unique aspects of the proposed method.

- Cost analysis. What's the cost of the Flow?  There is no discussion of Flow’s cost relative to other frameworks, leaving the efficiency of the proposed method unclear.

- Experiment specifications. What's the specified experiment settings? Important experimental settings, such as the number of candidate graphs (𝐾) and the agent count, are not specified.

- Updating machinism. The description of workflow refinement process lacks critical details. In line 318: the phrase “systematic review to determine if the workflow requires refinement” lacks clarity regarding how this review is conducted and the criteria for determining when refinements are needed. Additionally, further details are needed on “this rigorous verification process” mentioned in line 323, particularly regarding how the system handles verification when no errors are found.

- Flow basic execution. Lacks several key information of Flow deisgn and execution. 1) How Flow determines when all prerequisite tasks has been completed? 2) How Flow handle the cases where some parallel tasks have not yet finished --dose Flow wait, reprioritize tasks, or initiate partial updates in such scenarios? 3) Critical setup information is missing, such as the total number of agents available in Flow’s default configuration.

- The current experiments cover only three tasks, which may not be sufficient to substantiate Flow's superior performance comprehensively. Expanding the experiments to include more diverse tasks and standardized benchmarks would strengthen the claims regarding its generalizability and robustness.

- LLM dependency. The paper does not address how Flow would perform with less powerful LLMs, such as GPT-3.5 or other open-source models. Exploring this would provide valuable insights into the framework’s robustness and generalizability

[1] Qian C, Xie Z, Wang Y, et al. Scaling Large-Language-Model-based Multi-Agent Collaboration[J]. arXiv preprint arXiv:2406.07155, 2024.

[2] Zhuge M, Wang W, Kirsch L, et al. Language agents as optimizable graphs[J]. arXiv preprint arXiv:2402.16823, 2024.

[3] Hong S, Lin Y, Liu B, et al. Data interpreter: An llm agent for data science[J]. arXiv preprint arXiv:2402.18679, 2024.

[4] Liu Z, Zhang Y, Li P, et al. Dynamic llm-agent network: An llm-agent collaboration framework with agent team optimization[J]. arXiv preprint arXiv:2310.02170, 2023.

**Questions:**

Please refer to the questions in Weaknesses.

---

> ### Author Response · Authors · 2024-11-24
> **Response from Authors**
>
> **Q1. Directed acyclic graphs (DAGs) and graph-based frameworks have already been established as effective structures in LLM-based agent frameworks [1][2][3][4]. To strengthen its contribution, the paper should include a comparative analysis with these existing approaches, highlighting specific advantages and unique aspects} of the proposed method.**
>
> A1. Thank you very much for your constructive comments for help improve our paper and strengthen its contribution. We appreciate the opportunity to clarify the unique aspects and advantages of our framework compared to existing frameworks that utilize graph-based structures.
>
> Our framework, unlike existing models, is designed to encourage workflow modularity. This design choice not only facilitates high levels of parallelism but also minimizes dependency complexities between tasks. By quantitatively measuring and promoting modularity, our method enhances system efficiency, reduces potential bottlenecks, and decreases the likelihood of delays or failures associated with complex interdependencies in dynamic environments.
> Moreover, directly benefiting from this emphasis on modularity, our framework allows for highly flexible updates to the workflow during execution based on all generated data while maintaining system stability and robustness.
>
> To provide specific contrasts:
>
> - **DyLAN** (COLM'24) and **MACNET** (ICLR'25 submission) do not allow updates to the workflow post-deployment. Our model differs in that it not only permits but facilitates dynamic modifications during runtime.
>
> - **GPTSwarm** (ICML’24) updates the interactions between agents but not the topology of each agent. Our method, however, can also update the topology of each agent. As acknowledged in their paper, changing the topology is important for enhancing task planning.
>
> - **DataInterpreter** (ICLR'25 submission) revises workflows only after execution failures in subtasks, making adjustments to subsequent tasks without reconsidering previously completed tasks. Our method, in contrast, revises and changes all tasks based on globally generated information, addressing not only execution failures but also any deficiencies in achieving the tasks.
>
> We have revised the related work section to include the above. Thank you very much for helping to improve our paper.
>
> ---
>
>
> **Q2. Cost analysis. What's the cost of the Flow? There is no discussion of Flow’s cost relative to other frameworks, leaving the efficiency of the proposed method unclear.**
>
>
> A2. To quantitatively measure the cost of our framework, we used execution time as the standard. Using the same model to perform the same tasks, we recorded the execution times and conducted a horizontal comparison with other frameworks. Each task was executed five times, and the average execution time was calculated.
>
>
> |                 | Task               | Flow (w/o update) | Flow (w/ update) | MetaGPT          | CAMEL           | Autogen         |
> |-----------------|--------------------|-------------------|------------------|------------------|-----------------|-----------------|
> | **GPT-3.5-turbo** |                    |                   |                  |                  |                 |                 |
> |                 | Gobang Game        | 26.12 ± 11.35     | 33.57 ± 12.46    | 34.00 ± 15.12    | 121.52 ± 20.87  | 31.00 ± 14.67   |
> |                 | Conference Website | 23.46 ± 10.84     | 34.23 ± 13.12    | 85.14 ± 18.52    | 41.96 ± 12.89   | 44.00 ± 15.34   |
> |                 | Latex Beamer       | 18.34 ± 9.73      | 24.12 ± 10.89    | 29.92 ± 14.87    | 166.00 ± 22.64  | 31.00 ± 16.78   |
> | **GPT-4o-mini**  |                    |                   |                  |                  |                 |                 |
> |                 | Gobang Game        | 60.45 ± 14.78     | 72.34 ± 13.45    | 99.45 ± 16.92    | 110.94 ± 19.67  | 148.72 ± 25.34  |
> |                 | Conference Website | 22.78 ± 12.45     | 52.14 ± 14.89    | 127.49 ± 17.52   | 74.53 ± 18.34   | 86.78 ± 21.23   |
> |                 | Latex Beamer       | 44.21 ± 13.67     | 83.34 ± 15.89    | 66.72 ± 19.45    | 106.34 ± 20.78  | 95.21 ± 22.56   |

---

> ### Author Response · Authors · 2024-11-24
> **Response from Authors**
>
> **Q3. Updating machinism. How review is conducted and the criteria for determining when refinements are needed.**
>
>
> A3. By following your constructive comments, we have added the complete prompt for an update in our Appendix D.3.
>
> The update prompt is as follows:
>
> 1. **Update the Workflow**
>
>     - **Evaluate Completed Tasks**:
>         - **Focus**: Examine only tasks with `"status": "completed"`.
>         - **Check Data**:
>             - Ensure that `"data"` for each task is sufficient, detailed, and directly contributes to the `final_goal`.
>
>     - **Assess Workflow Structure**:
>         - **Examine All Tasks**: Review all tasks, including those labeled `"completed"`, `"pending"`, and `"in-progress"`.
>         - **Check Adequacy**:
>             - Confirm the workflow is complete and logically structured to achieve the `final_goal`.
>             - Ensure there are no missing critical tasks or dependencies.
>             - Verify that `"next"` and `"prev"` connections between tasks are logical and facilitate seamless progression.
>         - **Identify Inefficiencies**:
>             - Detect and address unnecessary dependencies, bottlenecks, or redundant steps that hinder the workflow's efficiency.
>
>     - **Allowed Changes**:
>         - **Modify**: Clarify and detail the objectives of tasks with insufficient or vague directives to ensure they meet the `final_goal`.
>         - **Add**: Introduce new tasks with clear, detailed descriptions to fill gaps in data or structure.
>         - **Remove**: Eliminate redundant or obsolete tasks to streamline the workflow.
>
>     - **Maintain Logical Flow**:
>         - Reorganize task connections (`"next"` and `"prev"`) to enhance parallel execution and improve overall workflow efficiency.
>
> 2. **Output Format**
>     - **If No Changes Are Made**:
>       - Return an empty JSON object to indicate that no modifications were necessary: `json
> {}`.
>     - **If Changes Are Made**:
>       - Return a JSON object containing the updated workflow without including the `"data"` fields to optimize token usage. This JSON should only include the structural changes (task parameters and connections).
>
> ### **An Example Input of workflow**:
> …
> …
> ### **Example Output Updated workflow**:
> …
> …
>
> ---
>
> **Q4. Updating machinism.  How Flow handle the cases where some parallel tasks have not yet finished --dose Flow wait, reprioritize tasks, or initiate partial updates in such scenarios?**
>
> A4. We implemented two different update strategies:
> 1. **Update Concurrently:**
> - In this approach, when a task is completed, it immediately triggers the workflow update function, even if other tasks are still running.
> - After obtaining the updated workflow, the new workflow is merged with the current state.
> - **Trade-off:** This workflow update strategy runs concurrently with task execution, optimizing running time. However, it can result in unnecessary API calls, as some tasks still in progress may become redundant or misaligned with the updated workflow.
> 2. **Lazy Update (Update After Task Completion) :**
> - In this strategy, when a task is completed, no new tasks are allocated immediately. Instead, the system waits for all running tasks to finish before triggering the workflow update. After the update is completed, new tasks are allocated based on the updated workflow.
> - This approach reduces unnecessary API calls by batching updates.
> - **Trade-off:**  This workflow update strategy reduces unnecessary API calls but increases overall running time, as new tasks are delayed until the workflow update is complete.
> In our paper, all the experiments are obtained by using the second strategy to avoid the waste of API usage.
>
> ---
>
> **Q5. Updating machinism.  How Flow determines when all prerequisite tasks have been completed?**
>
> A5. We check if the number of prerequisite tasks are completed.
>
> ---
>
>
> **Q6. Total number of agents available in Flow’s default configuration.**
>
> A6. By default, the number of agents is determined by the LLM during workflow generation, with the restriction that only one agent is assigned per subtask. Consequently, the maximum number of agents cannot exceed the total number of tasks in the workflow.
> This constraint can be easily adjusted to customize the prompt configuration if needed. During execution, the system dynamically checks the availability of agents to ensure that new subtasks are initiated only when the corresponding agents are ready.
>
> ---

---

> ### Author Response · Authors · 2024-11-24
> **Response from Authors**
>
> **Q7. Additionally, further details are needed on “this rigorous verification process” mentioned in line 323, particularly regarding how the system handles verification when no errors are found.**
>
> A7. The "rigorous verification process" refers to the careful validation steps undertaken to ensure the accuracy and integrity of the updated workflow. Since LLMs can sometimes produce errors, after obtaining the updated workflow and merging it with the latest version, we thoroughly review it to detect any inconsistencies or issues. We do not fully rely on fields such as `"status"` or `"num_parents_not_completed"` as generated by GPT. We also validate that the updated workflow forms a valid acyclic graph, ensuring logical task dependencies and preventing potential circular references.
>
> ---
>
>
> **Q8. Experiment specifications. What's the specified experiment settings? Important experimental settings, such as the number of candidate graphs ($K$) and the agent count, are not specified.**
>
> A8. Thank you for your question. In our paper, we set \( K = 10 \) which ensures that graphs generated for the same task across multiple runs have sufficient variation. Then generating process is parallel.  The number of agents is entirely generated by the workflow itself.
>
>
> ---
>
> **Q9. Expanding the experiments to include more diverse tasks and standardized benchmarks would strengthen the claims regarding its generalizability and robustness.**
>
> A9. Thank you for your question. We have added two new benchmarks across two different settings in the paper: 1). mathematical reasoning using GSM8K and 2). coding generation ability using MBPP. The results are presented as follows, all experiments using ChatGPT 4o-mini.
>
> | **Dataset** | **Direct** | **CoT** | **Ours** |
> |-------------|------------|---------|----------|
> | MBPP        | 57.7       | 58.2    | 76.4     |
> | GSM8K       | 92.7       | 92.4    | 93.0     |
>
>
>
>
>
> ---
>
>
> **Q10. LLM dependency. The paper does not address how Flow would perform with less powerful LLMs, such as GPT-3.5 or other open-source models. Exploring this would provide valuable insights into the framework’s robustness and generalizability**
>
> A10. We have followed your constructive comments and conducted experiments using GPT-3.5-turbo on three different tasks across various frameworks. Each task was executed five times to ensure consistency, and results were evaluated using the same scoring metrics described in Appendix B.5.
> From the results, we observe that even with a less powerful model like GPT-3.5-turbo, **Flow (Ours)** consistently outperforms other frameworks across various tasks. Notably, our framework demonstrates significant advantages in task quality, particularly for tasks requiring workflow management and decision-making capabilities.
> The results are shown in the tables below:
>
> **Gobang Game Performance**
>
> | **Model** | **Compilable (%)** | **Intractable (%)** | **Game Rule (%)** | **Overall Score (%)** |
> |-----------------------|--------------------|---------------------|-------------------|------------------------|
> | AutoGen  | 80 | 20 | 20 | 40 |
> | MetaGPT | 80 | 20 | 40 | 53 |
> | CAMEL  | 80 | 80 | 40 | 67 |
> | **Flow (Ours)** | **100** | **100** | **60** | **87** |
> ---
>
> **Website Design Performance**
>
> | **Model** | **Compilable (%)** | **Basic Information (%)** | **Sections (%)** | **Overall Score (%)** |
> |-----------------------|--------------------|---------------------------|------------------|------------------------|
> | AutoGen  | 20 | 0 | 0 | 7 |
> | MetaGPT  | 80 | 60 | **60** | 67 |
> | CAMEL  | 40 | 40 | 20 | 33 |
> | **Flow (Ours)** | **100** | **100** | 40 | **80** |
>
> ---
>
>  **LaTeX Beamer Writing Performance**
>
> | **Model** | **Compilable (%)** | **Completeness (%)** | **Page Limit (%)** | **Overall Score (%)** |
> |-----------------------|--------------------|-----------------------|--------------------|------------------------|
> | AutoGen  | 40 | 0 | 0 | 13 |
> | MetaGPT  | 20 | 20 | 0 | 13 |
> | CAMEL  | 80 | 80 | 0 | 53 |
> | **Flow (Ours)** | **100** | **100** | 0 | **67** |
> ---

---

> ### Author Response · Authors · 2024-11-25
>
> Dear Reviewer RcMF,
>
> Thank you for having taken your time to provide us with your valuable comments, which help improves our paper. This is a gentle reminder that the discussion period is nearing its conclusion, we hope you have taken the time to consider our responses to your review. If you have any additional questions or concerns, please let us know so we can resolve them before the discussion period concludes. If you feel our responses have satisfactorily addressed your concerns, it would be greatly appreciated if you could raise your score to show that the existing concerns have been addressed.
>
> Thank you!
>
> The Authors

---

> > ### Comment · Reviewer_RcMF · 2024-11-26
> >
> > Thank you for the authors' detailed responses. However, some concerns still remain:
> >
> > 1. Regarding time cost (Q2): Flow achieves shorter or comparable execution time despite generating multiple execution flows. Moreover, as mentioned in Q8, K=10, can you explain why the time cost is still low?
> >
> > 2. From Q3 and the revised version, the workflow update seems heavily reliant on prompt engineering. Have you applied any other techniques rather than prompts?
> >
> > 3. I found there is no ablation study, making it unclear how the important design choices affect the performance and the core module effectiveness.

---

> ### Author Response · Authors · 2024-11-30
> **Followup Response from Authors**
>
> **Dear Reviewer RcMF,**
>
> ### We apologize for sending our reply during Thanksgiving and the weekend.
>
> ### We have followed your constructive comments added many ablation studies. We believe these follow-up questions are important and has been carefully incorporated all results into our work. Thank you for your time to help us improve our paper. Please kindly let us know if your followup comments are addressed.
>
> ---
>
> **Followup Q1. Regarding time cost (Q2): Flow achieves shorter or comparable execution time despite generating multiple execution flows. Moreover, as mentioned in Q8, K=10, can you explain why the time cost is still low?**
>
> Followup A1. The key reason our system maintains a low time cost is:
> - Unlike existing graph-based systems, we explicitly encourage parallel execution and low dependency by calculating parallelism and dependency metrics for the generated workflows. By selecting multiple candidate graphs based on these quantitative metrics, we directly facilitate the creation of highly parallel and less complex workflows.
> - Although the work could be done in parallel, existing implementations typically do not execute tasks simultaneously but in sequence. `Our implementation fully utilizes parallel execution capabilities, significantly reducing the expected execution time. By benefiting directly from parallel processing, the 10 candidate graphs are generated simultaneously, so the expected running time is the same as generating a single graph.`
>
> To further illustrate this, following your suggestions, we have added an additional ablation study in `Followup A3 (Table 1)`. This study shows that without parallel execution, the effectiveness is approximately 2 to 3 times slower.
>
> ---
> **Followup Q2. From Q3 and the revised version, the workflow update seems heavily reliant on prompt engineering. Have you applied any other techniques rather than prompts?**
>
> Followup A2. Thank you for your insightful question. The prompt serves to establish a standard protocol that constrains the update rules, specifying what aspects should be checked and the required output format. The key techniques for achieving effective workflow update include 1). encouraging the modularity of workflow by using parallelism and low dependency metrics 2). implementing a lazy update strategy.
>
> 1). Specifically, as demonstrated in the ablation study in `Followup A3 (Tables 2 to 4)`, we measure the average proportion of subtasks indirectly affected (and requiring modification) by changes in a workflow using the Propagation Rate. This metric quantifies how effectively changes are localized within the workflow. The results show that `excluding the two key metrics—parallelism and low dependency—nearly doubles the Propagation Rate, leading to significantly more disruptions across the workflow during the update`. Workflows with high propagation rates are difficult to scale, as changes in one part of the system are more likely to cascade and disrupt other parts.
>
> To further illustrate the effectiveness of emphasizing modularity, we have also provided a theoretical result that modularized workflows have a high expected number of successful tasks (see `Followup A3 of Reviewer eHFH`).
>
> 2). Additionally, we have incorporated system-level implementation techniques. This would be helpful and provide a strong code foundation for future work of LLM multi agents. Specifically, we implement a `lazy update strategy`. In this strategy, no immediate updates or new task allocations are triggered upon the completion of an individual task. Instead, the system deliberately waits for all currently running tasks to finish before initiating a workflow update. Once all tasks have completed, the system performs the update in a single step and allocates new tasks based on the revised workflow. This strategy `reduce redundant updates and reduce waste of resources (API cost)`, which is important for designing a real-world multi-agent system in the future. We have also added an ablation study in `Followup A3 Table 5`  to demonstrate the effectiveness.
>
> We believe that our perspective and empirical analysis of the benefits of workflow modularization potentially can influence the design of future multi-agent systems. Our implementation such as parallelism executing and lazy updates provides a robust framework that can serve as a solid foundation for advancing research on multi-agent systems.

---

> > ### Author Response · Authors · 2024-12-01
> >
> > Dear Reviewer RcMF,
> >
> > We greatly appreciate your followup questions and have added ablation studies.  We have highlighted the important words in the followup response for your convenience. Thanks to your comments, the overall quality of our paper has been improved a lot.
> >
> > As the discussion period will soon close, could you kindly check whether your followup response have been addressed? If so, it would be greatly appreciated if you could adjust your score to reflect that the concerns have been addressed. If you have any additional questions or concerns, please let us know so we can resolve them before the discussion period concludes.
> >
> > Many thanks
> >
> > The authors

---

> > > ### Author Response · Authors · 2024-12-02
> > > **a kind reminder**
> > >
> > > Dear Reviewer RcMF,
> > >
> > > This is a kind reminder that we are still awaiting your response to our followup reply. Please kindly let us know if you have any further comments. We can still address in our paper.
> > >
> > > Best regards,
> > >
> > > The Authors

---

> > > > ### Comment · Area_Chair_bmJT · 2024-12-02
> > > >
> > > > Dear reviewer,
> > > >
> > > > May I know if the authors' response have addressed your concerns?
> > > >
> > > > Best,
> > > > AC

---

> > > > > ### Comment · Reviewer_RcMF · 2024-12-02
> > > > >
> > > > > Dear AC and Authors,
> > > > >
> > > > > Thank you for the additional results and the detailed clarifications provided in your response. After carefully reviewing the updated work and considering the improvements made, I believe my concerns regarding the ablation study and workflow update mechanisms have been addressed. As a result, I have decided to increase my score to 5.
> > > > >
> > > > > Best regards,
> > > > > Reviewer RcMF

---

> ### Author Response · Authors · 2024-11-30
> **Followup Response from Authors**
>
> **Followup Q3. I found there is no ablation study, making it unclear how the important design choices affect the performance and the core module effectiveness.**
>
> Followup A3. We follow your constructive comments and added additional ablation studies including 1) parallel execution, 2) with/without parallelism and dependency metrics, and 3) dffierent updating strategies.
>
>
> **1).** We examine the `effectiveness of parallel execution in reducing running time.`  Across all three cases, we use GPT-4o-mini, with the only difference being whether parallel execution is employed in implementation. As expected, we observe a significant reduction in the time needed to complete the same task.
>
>  ### Table 1: Comparison of Scores under Different Running Conditions
> |  Running Condition   |  Gobang time      |  Latex time       |  Website time      |
> |-------------------------|---------------------|---------------------|---------------------|
> |  Parallel Running     |  72.34 ± 13.45   |  52.14 ± 14.89    |  83.34 ± 15.89    |
> | No Parallel Running     | 176 ± 23.43        | 183 ± 25.81         | 210 ± 29.84         |
>
>  **2).** To explore the `effectiveness of parallelism and dependency metrics on the impact of subtask changes`, we conducted an ablation study with two settings: one using our validated metrics and the other without them.  We measures the average proportion of subtasks indirectly affected (and requiring modification) by changes in a workflow with the Propagation Rate .  Workflows with high propagation rates are hard to scale since changes in one part of the system are less likely to disrupt others.
> The Propagation Rate is defined as:
> $$
> \text{Propagation Rate} = \frac{1}{\text{Number of Updated Subtasks}} \sum \left( \frac{\text{Number of Indirectly Affected Subtasks}}{\text{Total Number of Subtasks in Workflow (per update)}} \right)
> $$
> 1. **Numerator of the Sum**: $\text{Number of Indirectly Affected Subtasks}$: The count of subtasks influenced as a result of a change in other tasks.
> 2. **Denominator of the Fraction**: $\text{Total Number of Subtasks in Workflow (Per Update)}$: The total number of subtasks in the workflow after the update, which serves as the baseline for normalization.
> 3. **Averaging Over Updated Subtasks**:  This ensures the Propagation Rate reflects the average number of indirectly affected subtasks per updated subtask.
>
>
>
>  ### Table 2: Impact of Dependencies on Workflow of Latex Beamer
> | Graph Type | Propagation Rate (%) |
> |----------------------|-----------------------|
> | With two metrics | 16 |
> | Without two metrics | 42 |
>  ### Table 3: Impact of Dependencies on Workflow of Gobang Game
> | Graph Type | Propagation Rate (%) |
> |----------------------|-----------------------|
> |With two metrics | 27 |
> | Without two metrics | 40 |
> ### Table 4: Impact of Dependencies on Workflow of Website Design
> | Graph Type | Propagation Rate (%) |
> |----------------------|-----------------------|
> | With two metrics | 34 |
> | Without two metrics | 73 |

---

> ### Author Response · Authors · 2024-11-30
> **Followup Response from Authors**
>
> **3).** To ablate our `update strategies for resource efficiency`, we conducted experiments under two different settings.
>
> - Update Concurrently:  In this approach, when a task is completed, it immediately triggers the workflow update function, even if other tasks are still running.
>
> - Lazy update:  In this strategy, no immediate updates or new task allocations are triggered upon the completion of an individual task. Instead, the system deliberately waits for all currently running tasks to finish before initiating a workflow update.
>
> We used two models: GPT-4o-mini and GPT-3.5-Turbo, for comparison. The experiment will involve the following metrics:
> 1. API Calls for Change: The total number of times the OpenAI API was called for additional workflow updates during the experiment.
> 2. Score: The average score across all three experimental settings, incorporating all relevant quantitative metrics.
> 3. Completion Time: The total time taken by the models to complete the tasks under both settings.
>
> we can see for both model Update After Task Completion resulting in the lower number of api calls per task and relatively the same accuracy rate also the completion time is shorter then Update Concurrently.  Additionally, The experiments show that  Update Concurrently strategy does not show significant improve performance. We think the reason is that for both methods, all subtasks will be eventually checked.
>
> ### Table 5: Number of  API Calls vs. Performance
> |  Strategy                             |  API Calls for Change  |  Overall Score (%)  |  Completion Time (s)      |
> |-----------------------------------------|---------------------------|-------------------|-----------------------------|
> | Update Concurrently (GPT- 4o-mini)           | 0.85                      | 95.57               | 66.75 ± 19.11               |
> | Update Concurrently (GPT-3.5-turbo)         | 0.92                      | 76.67               | 33.82 ± 13.45               |
> | Lazy Update  (GPT-4o-mini)  | 0.65                      | 93.33                | 69.27 ± 14.74           |
> | Lazy Update (GPT-3.5-turbo)| 0.68                      | 78.00          | 30.64 ± 12.16           |
>
> **4).** Additionally,  we have `simulated scenarios such as unstable internet connections or internal API errors (e.g., ChatGPT occasionally returning "there is an error, failed to generate a response")`. The results show that when a task's status becomes inconsistent with the generated data, our update mechanism effectively adds new, similar tasks while bypassing unnecessary steps associated with failed subtasks.
>
>
> Many thanks,
>
> Authors

---

> ### Author Response · Authors · 2024-12-02
> **Thanks, and kindly let us know if there have any concerns**
>
> Dear Reviewer RcMF,
>
> Thank you very much for your time and help us review our paper! We will follow your comments and carefully add these studies into our final version. Please kindly let us know if there are any other remaining concerns.
>
> Many thanks,
>
> Authors

---

### Official Review · Reviewer_Je7s · 2024-11-04

**Soundness:** 2
**Presentation:** 4
**Contribution:** 3
**Rating:** 6
**Confidence:** 3

**Summary:**

In this paper, the author(s) propose a multi-agent framework to enable dynamic workflow update, by integrating activity-on-vertex (AOV) graphs. In particular, an agent is prompted to generate several workflows in the form of AOV, and the one with the highest parallelism level and lowest dependency complexity will be selected. Various agent roles will be assigned to complete the workflow tasks, and the framework allows workflow refinement and dynamic updating if agents encounter errors.

**Strengths:**

- The paper adds value to agent-based workflow generation and automation.
- Activity-on-vertex graphs are incorporated to better manage task dependency and parallelism.
- Comparative analysis is conducted with multiple related work in terms of both success rate and human rating.

**Weaknesses:**

- A framework architecture or sequence diagram is needed to demonstrate the overview of the proposed solution.
- I am wondering why the proposed solution does not check the workflow correctness in the first place, but the parallelism and modularity.
- Section 4.1.3: Why “Flow gets 100% success rate” while in Table 3 the overall success rate is 80%?
- The evaluation results can be elaborated. For instance, in the three scenarios, how many updates are required respectively?
- Table 1: “task 1, 2 completed, task 3 under-work”, task 2 and 3 should be exchanged according to the above AOV.
- The paper organization can be adjusted. For instance, there is no Section 4.2, then why Section 4.1 is needed?
- A proof-reading is needed as some typos are found.

      - Section 2: “previous approach like …” -> “Previous”.

      - Section 4.1: “… the average performance of Flowis 93% …” -> “Flow is”.

**Questions:**

- I am wondering why the proposed solution does not check the workflow correctness in the first place, but the parallelism and modularity.
- Section 4.1.3: Why “Flow gets 100% success rate” while in Table 3 the overall success rate is 80%?
- The evaluation results can be elaborated. For instance, in the three scenarios, how many updates are required respectively?

**Details Of Ethics Concerns:**

Section 4.1: “We gathered 50 participants with programming and machine learning backgrounds to rank the outcomes produced by different methods.” Please confirm the author(s) have ethical approval for recruiting human participants.

---

> ### Author Response · Authors · 2024-11-24
> **Response from Authors**
>
> **Q1. why the proposed solution does not check the workflow correctness in the first place, but the parallelism and modularity?**
>
> A1. Thank you for your constructive comments. We realize that this point should be discussed more clearly in our paper. We have added the following discussion to the revised version from lines 261–271.
>
> We prioritize parallelism and modularity early in the process and focus on refining the workflow through data-driven adjustments during runtime. The reasons are as follows:
>
> 1). When leveraging LLMs to generate workflows for specific tasks, these models inherently possess reasoning capabilities that make the workflows reasonably reliable, even without explicitly emphasizing reliability in the prompts. However, the specific task can often be achieved through multiple workflows, many of which may not prioritize efficiency. If parallelism and independence are not explicitly encouraged during the initial workflow generation, the model might produce sequential or overly complex workflows, making them inefficient. Therefore, we emphasize parallelism and modularity from the outset.
>
> 2). We do not have additional data to verify correctness, and without such data, verifying correctness becomes inherently challenging. This is similar to the scientific process, where experimental validation and iterative refinements are necessary to improve the accuracy of physical laws. Since no supervised information is available at the beginning, we focus on refining the workflow during runtime as data becomes available.
>
> ---
>
> **Q2. The evaluation results can be elaborated. For instance, in the three scenarios, how many updates are required respectively?**
>
> A2. Based on your constructive comments, we have reported the updates in the following tables for all tasks using GPT-4o-mini. In this experiment, each task was run five times on different models, and the average of the results was calculated as the final outcome. Specifically, we evaluate the workflows using three metrics:
>
> - #Initial Tasks: The number of tasks identified within the workflow after selecting the optimal workflow but before execution begins.
>
> - #Changed Tasks: The number of tasks in the initial workflow that was updated after completing the workflow execution.
>
> - Changed Ratio: Calculated as the average number of changed tasks divided by the average number of initial tasks, providing an intuitive measure of the proportion of tasks updated during execution.
>
> We ran each task five times and reported the average values in the table below.
>
>
> | **LLM-agent** | **Task**            | **#Initial Task (avg.)** | **#Changed Tasks (avg.)** | **Changed Ratio (avg.)** |
> |---------------|---------------------|--------------------------|---------------------------|--------------------------|
> | GPT-4o-mini   | Gobang Game         | 8                        | 2.8                       | 35%                      |
> |               | Website Design      | 7.2                      | 3.4                       | 47%                      |
> |               | LaTeX Beamer Writing | 9.2                      | 4.8                       | 53%                      |
>
>
> Additionally, we have included a lower-performance model, GPT-3.5-turbo, in our evaluation. As expected, GPT-3.5-turbo required more updates during runtime as expected. This is because GPT-3.5-turbo has comparatively weaker task execution capabilities, resulting in more frequent workflow adjustments due to insufficiently generated data. Notably, our method outperformed the best baseline on GPT-3.5-turbo by **29.9\%**, **26.4\%**, and **19.2\%** on quantitative measures across the three tasks,  respectively (see Appendix B.5). This further highlights the effectiveness of our workflow generation and updating framework (Appendix B.4).
>
> | **LLM-Agent**   | **Task**            | **#Initial Tasks (avg.)** | **#Changed Tasks (avg.)** | **Changed Ratio (avg.)** |
> |-----------------|---------------------|---------------------------|---------------------------|--------------------------|
> | GPT-3.5-turbo   | Gobang Game         | 7.8                       | 3.4                       | 44%                      |
> |                 | Website Design      | 7.2                       | 4.8                       | 66%                      |
> |                 | LaTeX Beamer Writing | 6.2                       | 4.4                       | 71%                      |
>
>
> ---
>
> **Q3. The paper organization can be adjusted. For instance, if there is no Section 4.2, then why Section 4.1 is needed?**
>
> A3. We sincerely appreciate your detailed review, in the revised version, we have reorganized the subsection numbering in Section 4. In addition, we have rewritten this paper to some extent to ensure logical consistency.
>
> ---

---

> > ### Author Response · Authors · 2024-11-24
> > **Response from Authors**
> >
> > **Q4. Section 4.1.3: Why does “Flow get a 100\% success rate” while in Table 3 the overall success rate is 80\%?**
> >
> >
> > A4. This typo has been corrected to "80\% success rate" in the revised version. Our method demonstrates superior performance compared to the baselines on the task of website design. we have also carefully reviewed the manuscript and corrected other typos.
> >
> >
> > **Q5. Please confirm the author(s) have ethical approval for recruiting human participants.**
> >
> > A5. The ethics approval will be attached to the camera-ready version, adhering to the double-blind review process.
> >
> > ---
> >
> > **Q6. A framework architecture or sequence diagram is needed to demonstrate the overview of the proposed solution.**
> >
> > A6. Thank you for your valuable feedback. We appreciate your suggestion. We have included an overview of the framework architecture in Appendix E of the paper. Due to its length, we have omitted it from this response.

---

> ### Author Response · Authors · 2024-11-25
>
> Dear Reviewer Je7s,
>
> Thank you for your time and review of our paper! We have followed your constructive comments and added additional experiments for cost analysis. Additionally, we have also included a lower performance model, GPT-3.5. If you have any further questions or concerns, please let us know so we can resolve them before the discussion period concludes. If you feel our responses have satisfactorily addressed your concerns, we would greatly appreciate it if you could raise your score to reflect that the issues have been addressed.
>
> Thank you!
>
> The Authors

---

> > ### Comment · Reviewer_Je7s · 2024-11-26
> > **Official Comment by Reviewer Je7s**
> >
> > I thank the authors for addressing the previous comments. Based on the revised manuscript I changed my rating to 6.
> > I suggest the authors to conduct further review as there are still some typos. For example, Section 4.3: "Experimental results are presented in Table ?? with explanations as follows".

---

> > > ### Author Response · Authors · 2024-11-30
> > >
> > > Dear Reviewer Je7s,
> > >
> > > You are so kind! Thank you for your efforts and detailed review. We truly appreciate your detailed feedback and will carefully review the manuscript to incorporate all your positive comments in the future version.
> > >
> > > Have a great weekend, and all the best wishes for a wonderful Thanksgiving!
> > >
> > > Authors

---

### Author Response · Authors · 2024-11-24

**Dear Reviewers**,

We sincerely thank you for your thorough and insightful feedback, which has significantly enhanced our paper. We have carefully addressed all your comments and made substantial revisions accordingly. Below, we provide a brief summary for your convenience.

1. **Clarification on Workflow Correctness (Reviewer Je7s)**: We have added a detailed discussion (lines 261–271) explaining why we prioritize parallelism and modularity over initial workflow correctness. Specifically, we emphasize that Large Language Models (LLMs) inherently produce reasonably reliable workflows due to their reasoning capabilities. Verifying correctness without additional data is challenging; hence, we focus on refining the workflow during runtime as data becomes available.
2. **Expanded Evaluation and Results (Reviewers Je7s, Reviewer RcMF,  Reviewer uiek,   Reviewer eHFH)**: We have enriched our experimental section by including standard benchmarks such as GSM8K for mathematical reasoning and MBPP for coding tasks. The results demonstrate that our framework outperforms baseline methods, even when using less powerful models like GPT-3.5-turbo. We have also provided detailed metrics on the number of updates required during task execution (Appendix B.4), which highlights the effectiveness of our dynamic workflow updating mechanism.
3. **Paper Organization and Clarity (Reviewers Je7s, Reviewer uiek)**: We have reorganized the paper for better logical flow, corrected typos, and improved grammar throughout. Figures, tables, and section numbering have been updated for consistency and clarity.
4. **Comparative Analysis and Technical Novelty (Reviewers RcMF, Reviewer eHFH)**: We expanded the related work section to include a comparative analysis with existing frameworks utilizing graph-based structures. We highlighted the unique aspects of our approach, such as our emphasis on workflow modularity and dynamic and global restructuring capabilities, which distinguish us from prior work.
5. **Cost Analysis and Experimental Settings (Reviewer RcMF)**: We included a detailed cost analysis using execution time as a metric, demonstrating the efficiency of our method compared to other frameworks.
6. **Dynamic Workflow Restructuring Mechanism (Reviewers RcMF, Reviewer eHFH)**: We provided comprehensive details about our workflow update strategies and the rigorous verification process in Appendix D.3. This includes how we handle unsolvable subtasks, adjust dependencies, and ensure continuity in task execution, thereby enhancing system robustness.

We believe these revisions comprehensively address your concerns and enhance the overall quality and clarity of our paper. Your constructive feedback was invaluable, and we hope our responses meet your expectations.

**Sincerely,**

The Authors

---

### Meta-Review · Area_Chair_bmJT · 2024-12-18

**Metareview:**

The paper presents a novel framework that integrates Activity-on-Vertex (AOV) graphs into large language models, aiming to enhance modularity and enable dynamic workflow updates. The empirical results indicate performance improvements in diverse tasks such as coding, game development, and website design. The authors have added in-depth ablation studies and theoretical analyses to underscore the advantages of their approach.

Strengths of this paper include addressing the key issue of modularity in multi-agent systems and improving the clarity and depth of the evaluation. The paper also displays a commendable effort in expanding upon theoretical justifications post-rebuttal, which the reviewers appreciated.

However, the paper faced critiques regarding the novelty of the approach and the robustness of evaluation criteria, particularly concerning the reliability of human evaluations. While the authors addressed these points in their rebuttal, not all concerns were fully resolved.

The decision to accept the paper rests on its substantial contributions to the domain of multi-agent systems, especially in the context of large language models. The authors' responses during the review process and their efforts to enhance the paper significantly influenced this positive outcome.

**Additional Comments On Reviewer Discussion:**

During the discussion, Reviewer RcMF questioned the novelty of the approach and the choice of metrics, while Reviewer Je7s expressed concerns about the evaluation methods and benchmarks. In their rebuttal, the authors provided a strong justification for their metrics and broadened their evaluation with standard benchmarks, addressing the issues raised by Je7s and partially resolving RcMF's concerns regarding novelty. They also detailed the mechanisms for workflow updates and presented additional ablation studies, which clarified implementation details and addressed concerns about resource efficiency raised by Je7s and uiek.

---

### Decision · Program_Chairs · 2025-01-22

Accept (Poster)